

**Smoke in the river: an AEROCLO-sA case study**
Cyrille Flamant[1], Marco Gaetani[1,2,3], Jean-Pierre Chaboureau[4], Patrick Chazette[5], Juan Cuesta[2], Stuart
J. Piketh[6], and Paola Formenti[2]
[1]Laboratoire Atmosphère Milieux Observations Spatiales (LATMOS)/IPSL, UMR CNRS 8190, Sorbonne
Université, UVSQ, Paris, France
[2]Université de Paris and Univ Paris Est Creteil, CNRS, LISA, F-75013 Paris, France
[3] Scuola Universitaria Superiore IUSS, Pavia, Italy
[4]Laboratoire d'Aérologie (LAERO), UMR CNRS 5560, Université de Toulouse, Toulouse, France
[5]Laboratoire des Sciences du Climat et de l'Environnement (LSCE)/IPSL, UMR CNRS 1572, CEA, UVSQ,
Gif-sur-Yvette, France
[6]School of Geo- and Spatial Science, North-West University, Potchefstroom, South Africa
Correspondence to: Cyrille Flamant (cyrille.flamant@latmos.ipsl.fr)
**Abstract**
The formation of a river of smoke crossing southern Africa is investigated during the Aerosols,
Radiation and Clouds in southern Africa (AEROCLO-sA) campaign in September 2017. A complementary
set of global and mesoscale numerical simulations as well as ground-based, airborne and space-borne
observations of the dynamics, thermodynamics and composition of the atmosphere are used to
characterize the river of smoke in terms of timing and vertical extent of the biomass burning aerosol
(BBA) layer.
The study area was under the synoptic influence of a coastal low rooted in a tropical easterly wave, a
high-pressure system over the continent and westerly waves in mid-latitudes, one of which had an
embedded cut-off low (CoL). The coastal low interacted with the second of two approaching westerly
waves and ultimately formed a mid-level temperate tropical trough (TTT). The TTT created the fast
moving air mass transported to the southwestern Indian Ocean as a river of smoke. The CoL, which
developed and intensified in the upper levels associated with the first (easternmost) westerly wave,
remained stationary above northern Namibia prior to the formation of the TTT and was responsible
for the thickening of the BBA layer.
This shows that the evolution of the river of smoke is very much tied to the evolution of the TTT while
its vertical extent is related to the presence of the CoL. The mechanisms by which the CoL, observed
over Namibia in the entrance region of the river of smoke, influences the vertical structure of the BBA
layer is mainly associated with the ascending motion above the BBA layer. In the presence of the CoL,
the top of the BBA layer over northern Namibia reaches altitudes above 8 km. This is much higher than
the average height of the top of the BBA layer over the regions where the smoke comes from (Angola,
Zambia, Zimbabwe, Mozambique) which is 5 to 6 km.
The results suggest that the interaction between the TTTs and the CoLs which form during the winter
may have a role in promoting the transport of BBA from fire-prone regions in the tropical band to the
temperate mid-latitudes and southwestern Indian Ocean.

**Key words:** Biomass burning aerosols; southern Africa fires; cut-off low; tropical temperate trough;
planetary boundary layer; Meso-NH; ERA5; CAMS; airborne lidar; CATS; MODIS; AERONET



## 1. Introduction

Widespread, coherent bands of smoke from forest fires are regularly observed to cross the southern African sub-continent near the end of southern Africa's dry season, and particularly in September (Annegarn et al., 2002; McMillan et al., 2003; Swap et al., 2003). These features are generally referred to as 'rivers of smoke' (owing to the sharply defined boundaries of the smoke plume, giving them the appearance of a 'river') and can be several hundred kilometres wide and extend over a few thousands kilometres while flowing off the southeast coast of Africa. The smoke transported in these rivers is coming from thousands of agricultural fires as well as accidental forest fires burning in Angola, Zambia, Zimbabwe, Mozambique, the Democratic Republic of Congo and South Africa, favoured by dry conditions during the austral winter (see Figure 1 in Roberts et al., 2009). The smoke from the southern African sub-continent is generally contained in multiple stratified layers trapped below approximately 850hPa, 700hPa and 500hPa, depending on synoptic conditions (Stein et al., 2003). The river-of-smoke events generally correspond to the direct eastward transport of biomass burning aerosols (BBA) from southern Africa to the southwest Indian Ocean in five main transport paths classified by Garstang et al. (1996). These trajectories lead to the transport of massive amounts of aerosols and gases (e.g. carbon monoxide) towards the southwest Indian Ocean (Duflot et al., 2010) and as far as southeast Australia (Pak et al., 2003; Sinha et al., 2004), with potential important implications for the radiative budget and the marine productivity of the region (Luo et al., 2008).

The aerosol properties, transport and distribution across southern Africa, the South Atlantic and the Indian Ocean have been widely investigated for their key role in controlling the radiative budget in the region and, consequently, global climate dynamics (Zuidema et al. 2016a, b; Haywood et al. 2021; Redemann et al. 2021). Recently, the long term aerosol transport in the region has been characterised in terms of synoptic weather patterns (Gaetani et al., 2021). Nevertheless, to date, the rivers of smoke have exclusively been studied in the framework of the 2000 Southern African Regional Science Initiative (SAFARI 2000) using ground-based, airborne and space-borne observations (Annegarn et al., 2002; Jury and Freiman, 2002; Swap et al., 2003; McMillan et al., 2003; Stein et al., 2003; Schmid et al., 2003; Pak et al., 2003; Magi et al., 2004) as well as numerical simulations (Sinha et al., 2004). Among other studies, Stein et al. (2003) have shown that they form in regions where high-pressure and low-pressure synoptic-scale systems are juxtaposed to constrain the shape of the aerosol rivers.

Rivers of smoke are very effective at transporting large amounts of smoke from the fire-prone tropical regions into the mid-latitude westerly circulation (Annegarn et al., 2002). Surface synoptic conditions in the tropical zone are generally dominated by easterly waves and associated lows, whereas aloft a more stable anticyclonic circulation prevails in association with the continental high pressure system. The subtropical latitudes are dominated by continental highs and baroclinic westerly waves at all levels (Jury and Freiman, 2002). Hence, the transport of smoke from the Tropical Africa to the southwest Indian Ocean implies the formation of complex synoptic systems that can bridge two climatic regions, namely the tropical band (10–20°S) and the temperate subtropical band (20-30°S) over the southern Africa sub-continent.

Tropical temperate troughs (TTTs) are generally described as an interaction between tropical convection and mid-latitude transient perturbations (see Todd et al. 2004 and references therein) and typically form when a tropical disturbance in the lower atmosphere is coupled with a mid-latitude trough in the upper atmosphere (Lyons, 1991). Mid-latitude baroclinic waves are a necessary condition for TTT development (Macron et al., 2014). TTTs are known to be the dominant rainfall-producing weather system over southern Africa during the austral summer, when they form a cloud band that extends along the northwest-southeast direction across the landmass and the adjacent southwest Indian Ocean (Ratna et al., 2013; Howard et al., 2019; and references therein) and tend to propagate eastward. TTT events have been suggested to act as a major mechanism of poleward transfer of moisture owing to the strong convergence forming a pronounced poleward flux along the cloud band (Ratna et al., 2013). However, the role of TTTs in the transport of BBA during the winter has never been investigated until now.



During the AEROCLO-sA (AErosol, RadiatiOn and CLOuds in southern Africa) field campaign held in
Namibia in August-September 2017 (Formenti et al., 2019), a river of smoke was observed to sweep
through southern Africa from west to east, between 5 and 6 September 2017. The study area was
under the synoptic influence of a coastal low rooted in a tropical easterly wave, a high-pressure system
over the continent and westerly waves in the mid-latitudes, one of which had an embedded cut-off
low (CoL; Favre et al., 2012). During this period, the coastal low interacted with the second of two
approaching westerly waves and ultimately formed a mid-level TTT, which created the fast moving
limb of air transported to the southwest Indian Ocean as a river of smoke. The CoL, that developed
and intensified in the upper levels associated with the first (easternmost) of the two approaching
westerly waves, remained stationary above northern Namibia prior to the formation of the TTT (from
2 to 4 September 2017) and was responsible for the thickening of the BBA layer that subsequently was
conveyed southeastward. The objective of the paper is to assess the respective impact of both the CoL
and the TTT on the atmospheric circulation and composition in the mid- and lower troposphere over
southern Africa prior to and during the river of smoke event.
In Section 2 we present the model simulations and measurements used in the study. In Section 3, we
detail the life cycle of the CoL and the TTL over Western Namibia, while their impact on the formation
of the river of smoke event over southern Africa are analysed in Section 4. We also detail the vertical
distribution and origin of smoke in the lower troposphere over northern Namibia, in the entrance
region of the river of smoke, using airborne measurements made on 5 and 6 September. In Section 5,
the impact of the CoL and the TTT on the tropospheric composition over the sub-continent is
investigated. Finally, we summarize and conclude in Section 6.

**2. Data**
2.1 Modelling
*2.1.1 ECMWF reanalysis products: ERA5 and CAMS*
The regional circulation over continental southern Africa and adjacent oceans for the period 1 – 6
September 2017 is analysed using ERA 5 reanalysis (Fifth European Centre for Medium-range Weather
Forecast Reanalysis; Hersbach et al. 2018). The reanalysis outputs are available every hour on a 0.25°
x 0.25° grid, as well as 137 pressure levels, 88 of which are below 20 km and 60 below 5 km (note that
only 37 levels are available for download). Among the available variables, we focus on mean sea level
pressure (MSLP), geopotential height (Z), potential vorticity (PV), outgoing longwave radiation (OLR),
vertical velocity ($\omega$) and wind.
We also make use of the Copernicus Atmospheric Monitoring Service (CAMS; Inness et al., 2019)
reanalysis available every 3 hours (00, 03, 06, 09, 12, 15, 18 and 21 UTC) with a resolution of
approximately 80 km and 60 pressure levels (37 of which are below 20 km and 20 below 5 km), to
analyse the evolution of BBA during the episode of interest (note that only 25 levels are available for
download). For consistency with ERA5 data, CAMS data have been interpolated onto a 0.25° regular
grid. In the following, we use organic matter as a proxy for BBA. In addition, we have compared to
same dynamical and thermodynamic variables as for ERA5 in order to check the consistency between
the two types of products. This is essential as we intend to investigate the relationship between the
atmospheric dynamics and the distribution of BBA. It turns out that fields such as PV, $\omega$, wind, relative
humidity are very consistent between ERA5 and CAMS (not shown).
The evolution of the BBA transport and the associated atmospheric circulation from 1 to 6 September
2017 above Namibia is analysed by using a principal component analysis (PCA) of organic matter
aerosol optical thickness (AOT) at 550 nm, MSLP and Z at 700 and 300 hPa from the CAMS reanalysis
product. PCA consists in projecting data variability onto an orthogonal basis by solving the eigenvalue
problem of the data covariance matrix, so that data variability is decomposed into independent
variability modes, each explaining a fraction of the total variability (von Storch and Zwiers, 1999). Each


variability mode is presented as an empirical orthogonal function (EOF), accounting for the anomaly
pattern of the variable related to the mean of the analysed period, and the associated PC time series
accounting for the evolution of the anomaly amplitude. It follows that negative values indicate a
reversal of the anomaly pattern.

### 2.1.2 Meso-NH

A high-resolution simulation of the atmospheric dynamics, thermodynamics and composition for the
case study was also run with the non-hydrostatic mesoscale model Meso-NH (Lac et al. 2018), version
5.4, over a domain covering southern Africa. The model was run on a grid with 5-km horizontal spacing
and 64 levels with a resolution of 60 m close to the surface to 600 m at high altitude. It was run for 6
days starting from 000 UTC on 1 September 2017. The lateral boundary conditions were given by
ECMWF operational analysis. The simulation used the Surface Externalisée (SURFEX) scheme for
surface fluxes (Masson et al. 2013), a 1.5-order closure scheme for turbulence (Cuxart et al. 2000), an
eddy-diffusivity mass-flux scheme for shallow convection (Pergaud et al. 2009), a microphysical
scheme for mixed-phase clouds (Pinty and Jabouille 1998), a statistical scheme for subgrid cloud cover
(Chaboureau and Bechtold 2002), the Rapid Radiative Transfer Model (Mlawer et al. 1997) for
longwave radiation and the two-stream scheme (Fouquart and Bonnel 1986) for shortwave radiation.
Emission, transport and deposition of dust were parameterized using a prognostic scheme (Grini et al.
2006) to allow online interaction with radiation. A black carbon passive tracer was emitted following
the daily Global Fire Emissions Database (GFED) version 4, available at a horizontal resolution of
0.25°x0.25° (van der Werf et al. 2017). Backward trajectories were computed using three passive
tracers initialized with the 3D coordinates of each grid cell at their initial location (Gheusi and Stein
2002). A mass extinction efficiency of 5.05 $m^2\,g^{-1}$ (representative of aged smoke as in Mallet et al.,
2019) was used to compute AOT at 550 nm from black carbon concentration simulated by Meso-NH.

### 2.2 Observations

### 2.2.1 Ground-based observations

The National Aeronautics and Space Administration Aerosol Robotic Network (NASA AERONET)
operates a number of sun spectral photometers in Namibia and South Africa, providing long-term and
continuous monitoring of aerosol optical, microphysical and radiative properties. We use level-2.0
(cloud-screened and quality-assured) AOT at 500 nm data. AERONET stations of interest are located in
Windpoort, Bonanza, the HESS (High Energy Stereoscopic System) site and Henties Bay (Namibia) as
well as Upington (South Africa), (see **Figure 1**).

### 2.2.2 Airborne observations

For the period from 5 to 12 September 2017, dynamics and thermodynamics profiles over continental
Namibia were obtained from dropsondes released from a high-flying aircraft, the French Falcon 20
aircraft for environmental research of Safire (Service des Avions Français Instrumentés pour la
Recherche en Environnement) based in Walvis Bay, on the west coast of Namibia (see **Figure 1**). The
vertical structure of the aerosol layers was obtained from the nadir-pointing airborne lidar LEANDRE
Nouvelle Génération (LNG; Bruneau et al., 2015) installed on the same aircraft. Details about the Safire
Falcon 20 flights, the lidar LNG and the dropsonde launching unit can be found in Formenti et al. (2019).
In the following we shall only analyse lidar and dropsonde data acquired on 5 and 6 September 2017.
On 5 September, the Falcon 20 circuit from and to Walvis Bay was performed counter clockwise from
0736 to 1014 UTC (flight F06). On 6 September, the Falcon 20 circuit was performed clockwise from
1055 to 1401 UTC (flight F09). On both flights, the aircraft flew around 10 km above mean sea level
(AMSL), except on 6 September between 1145 and 1218 UTC when the aircraft performed a sounding
and penetrated the BBA layer over the Etosha pan. In the following, we will discuss the dynamics and
thermodynamics profiles from 2 dropsondes released over the Etosha pan at nearly the same location:
16.445°E / 18.772°S on 5 September at 0839 UTC and 16.401°E / 18.766°S on 6 September at 1146
UTC. We also compare these profiles to the one obtained in the vicinity of Henties Bay, over the ocean




at 13.78°E / 21.69°S on 5 September at 0951 UTC, and to those obtained south of Etosha at 16.33°E /
21.74°S on 6 September at 1137 UTC. Note that the drosponde data acquired during AEROCLO-sA have
not been assimilated in the ECMWF operational analysis nor in the reanalysis.
The signal backscattered to the LNG system telescope at 1064 nm is range-square-corrected to
produce atmospheric reflectivity. Total attenuated backscatter coefficient (ABC) profiles are derived
from atmospheric reflectivity profiles by normalizing the atmospheric reflectivity above the aerosol
layers to the molecular backscatter coefficient profiles. Hence the slope of the lidar reflectivity above
7.5 km AMSL matched that of the molecular backscatter derived from dropsonde measurements of
pressure and temperature. In the following, we only use ABC at 1064 nm because the attenuation by
the BBA in the lower troposphere, in spite of being important, does not prevent the lidar signal from
reaching the surface, unlike at the other wavelengths of operation of LNG (i.e. 355 and 532 nm). The
vertical resolution of the ABC profiles is 30 m. Profiles are averaged over 5 s, yielding a horizontal
resolution of 1 km for an aircraft flying at 200 m s$^{-1}$ on average. It is worth noting that ABC as observed
with LNG is sensitive to both aerosol concentration and aerosol hygroscopicity. Indeed, relative (RH)
in excess of 60% modify the size and the complex refractive index of aerosol, and hence their optical
properties, enhancing the ABC (e.g. Randriamiarisoa et al., 2006).
*2.2.3 Space-borne observations*
We make use of AOT (at 550 nm) and fire hot-spot locations from the National Aeronautics and Space
Administration Moderate Resolution Imaging Spectroradiometer (NASA MODIS; King et al., 1992). We
also make use of ABC and aerosol typing products obtained from the space-borne Cloud-Aerosol
Transport System (CATS; Yorks et al., 2016) to gather information on the vertical structure of aerosol
and cloud layers as well as aerosol composition over Namibia from two overpasses on 4 and 5
September 2017. Details about the space-borne products used in this study can be found in Chazette
et al. (2019). The horizontal distribution of the smoke plumes is also described daily with total column
amounts of carbon monoxide (CO) derived from radiance spectra measured by the Infrared
Atmospheric Sounding Spectrometer (IASI; Clerbaux et al., 2009), on-board the MetOp-A and MetOp-
B satellites and overpassing the region around 0830 and 0900 LT (local time) respectively. This satellite
dataset is retrieved using the Fast Optimal Retrievals on Layers for IASI algorithm (Hurtmans et al.,
2012) and validated against airborne and space-borne observations (Georges et al., 2009; De Watcher
et al., 2012). Finally, we utilize RGB natural colour imagery of cloud cover obtained with the Spinning
Enhanced Visible and InfraRed Imager (SEVIRI) instrument on-board the geostationary satellite
MeteoSat Second Generation.

**3. Synoptic conditions over southern Africa: a tale of two features**
3.1 The life cycle of the CoL and the TTT
In the low levels, the northern part of Namibia is under the influence of easterly flow, and on 1
September a marked coastal trough is seen along the west coast of Namibia, west of the instrumented
sites, associated with an easterly wave (**Figure 2a**). In the following days (2-3 September), a weak
coastal trough is seen to develop offshore (**Figure 2b,c**). Further south, a belt of high pressure systems
is present in the vicinity of the tip of the southern Africa sub-continent (e.g. **Figure 2a-c**). From 4
September, as the baroclinic westerly flow approaches the west coast, a distinct low forms over Angola
and northern Namibia within the easterly flow (**Figure 2d-f**), which corresponds to the Angola low in
the mid-troposphere. During the same period, the Southeast Atlantic high pressure system (St Helena
anticyclone) is strengthening (**Figure 2d**). Its eastern fringe is approaching the land on 5 September
(**Figure 2e**) and even intrudes over the sub-continent on 6 September (**Figure 2f**).
**Figure 3** shows the evolution of the ERA5 geopotential and potential velocity (PV) at 300 hPa at 1200
UTC over the area of interest. On 1 September, the instrumented sites over northern Namibia are
under the influence of a westerly flow in the upper levels, which is clearly separated from the main


westerlies located poleward of 40°S (**Figure 3a**). The split in the westerlies is generally associated with
the breaking of the upper level jet and seen upstream of southern Africa at 10°W (Favre et al., 2012).
Strong negative PV is associated with the cyclonic circulation in the area of separation between the
main westerlies and the northern westerlies branch, just west of the Namibian coastline. On 2
September, the region of splitting reaches the coastline, and some of the instrumented sites in the
southern part of the domain and along the coast are under the influence of a southerly flow, while the
sites to the north are under a westerly flow (**Figure 3b**). The maximum of negative PV is located
approximately above the AERONET HESS site. The main westerly flow exhibits a pronounced poleward
dip just west of the tip of South Africa which is further enhanced on the following day, when the closed
circulation has formed and is centred over the region of the AERONET Bonanza site (**Figure 3c**). This
closed circulation now contains the strongest negative PV feature. In the ERA5 reanalysis, ascending
motion is present to the north and east of the CoL centre, while descending motion is highlighted to
the south and west of the CoL centre (**Figure S1a**). The CoL is positioned in the same area the next day
(4 September, **Figure 3d**), with the instrumented sites being essentially beneath the negative PV
associated with the CoL. On this day also, ascending motion is present to the north and east of the CoL
centre, while descending motion is highlighted to the south and west of the CoL centre (**Figure S1b**).
The CoL starts deforming shortly after and becomes elongated in the north-south direction, and can
be seen as an elongated filament of negative PV to the east of most of the instrumented sites located
in Namibia. It is then moving poleward and merges back with the main westerly flow between 0900
and 1200 UTC on 5 September (not shown). The area of interest lies below a hump in the north-
westerly flow on the eastern side of the poleward dip of the westerlies (**Figure 3e**). Finally, on 6
September, the area of interest in under a rather weak east-northeasterly flow, north of the main
westerlies located south of 40°S (**Figure 3f**). The filament of negative PV marking the remains of the
CoL has now moved further southeast over South Africa and over the Indian Ocean.
On 3 September, the CAMS-derived circulation at 700 hPa shows, equatorward of the strong
westerlies, the presence of a high pressure system over the southern tip of the continent as well as an
isolated low pressure feature (connected with the CoL) located offshore of Namibia, over the Atlantic
Ocean (**Figure 4a**). The cyclonic and anticyclonic circulations associated with the CoL and the high
pressure, respectively, create conditions that are favourable to the advection of BBA poleward,
between them as shown by Chazette et al. (2019). It is worth noting that at this level, both the low and
high pressure features are characterised by low BBA-related AOT. As the wave in the westerly flow at
700 hPa approaches the west coast of southern Africa on 4 September, the high pressure moves
eastward and is partly over the Indian Ocean (**Figure 4b**). At the same time, the low pressure becomes
elongated and is oriented almost parallel to the coastline (a shape that resembles that observed at 300
hPa on 4 September, see **Figure 3c**). The poleward advection of BBA at 700 hPa becomes more
pronounced and results in the formation of the river of smoke. On 5 September (**Figure 4c**), the
equatorward dip of the westerly flow is now positioned over southern Africa and the high pressure has
moved over the Indian Ocean (~20°W), generating a TTT aligned with the Namibian coast. The resulting
circulation promotes the poleward advection of a more massive river of smoke extending over the
eastern coast of southern Africa and over the Indian Ocean, and extends poleward almost to 50°S. The
signature of the CoL at 700 hPa is no longer identifiable. BBA over the Atlantic Ocean related to biomass
burning events over South America are also observed by Chazette et al. (2019). Finally, on 6 September,
the St Helena anticyclone pushes the poleward dip of the westerlies and the TTT further east, while
the high pressure system over the Indian Ocean also weakens (**Figure 4d**). The river of smoke is now
well established over southern Africa, and located further east, particularly over South Africa.
3.2 Dynamical controls
The variability of the BBA distribution in the period 1-6 September is compared with the atmospheric
circulation variability by means of a PCA. The first EOF of the organic matter AOT (**Figure 5a**), explaining
55% of the variability, shows a negative anomaly in the BBA transport until 4 September and a river of
smoke developing over Namibia on 5 and 6 September (red line in **Figure 5e**). BBA are mainly



transported at 700 hPa, and the evolution of the river of smoke is controlled by the circulation at this
level. The first EOF of the geopotential at 700 hPa (**Figure 5c**), explaining 74% of the variability at this
level, shows on 1-2 September a positive gradient along the northeast-southwest direction, which
drives a southeasterly flow inhibiting the northerly BBA transport. The gradient reverses on 4
September (green line in **Figure 5e**), favouring the southward penetration of the river of smoke. The
control on the BBA exerted by the circulation at 700 hPa is confirmed by the high correlation (0.78,
**Table 1**) of PCA time series in **Figure 5e**. The first EOF of the geopotential at 300 hPa (**Figure 5b**),
explaining 56% of the variability, shows the transit of the CoL along the meridional direction, migrating
from southern Namibia on 1 September to central Namibia on 3-4 September and retreating to the
south on 5-6 September (blue line in **Figure 5e**). The evolution of the AOT and geopotential PCAs
suggests a possible control of the river of smoke by the CoL from 4 September onwards (PCA
correlation is 0.50, **Table 1**), when high-low pressure dipole (**Figure 5b**) favours the channelling of the
BBA along the Namibian coast. The first EOF of the MSLP (**Figure 5d**), explaining 43% of the variability,
shows a coastal trough on 1-2 September evolving into a coastal ridge on 3-4 September (black line in
**Figure 5e**). The associated northwesterly flow could have a role in favouring the installation of the river
of smoke. However, the comparison of the PCAs (**Table 1**) shows no stable relationship between the
conditions at the surface and the BBA transport. In conclusion, the CoL, and the circulation at 300 hPa,
has a dominating influence on the circulation around 700 hPa (~4 km AMSL) and consequently on the
tropospheric composition over Namibia, as the circulation at this level controls the distribution of BBA
away from the main sources.

**4. Smoke and clouds over southern Africa**
4.1 The river of smoke in satellite and ground based observations
The position of the river of smoke in the CAMS reanalysis at 1200 UTC in **Figure 4b-d** matches the
observations retrieved from MODIS (**Figure 6**) and IASI (**Figure S2**), respectively in terms of AOT and
CO total amounts, on 4, 5 and 6 September. In particular, we remark the southward progression of the
plume between 4 and 5 September (**Figures 6a,c and S2b,c**) and then its eastward displacement
reaching the continent on 6 September (**Figures 6c, e and S2c,d**). For the AOT measurements, the
signature of the river of smoke is more distinct on 6 September over the continent (**Figure 6e**), thanks
to the mostly cloud-free conditions over southern Africa on that day. On the other hand, the river of
smoke on 5 September is more difficult to detect with the MODIS observations (**Figure 6c**), due to its
co-location with a band of mid-level clouds positioned along the Namibian coastline. CO retrievals
depicts the river of smoke both on 5 and 6 September (**Figure S2c,d**), as these measurements can also
be retrieved above low and mid-level relatively thin clouds. On 3-4 September, enhanced CO amounts
(adjacent to clouds, shown as blanks in the figure) already show an elongated plume over the Atlantic
extending from northwest to southeast over the latitude band 10-40°S, that probably corresponds to
the river of smoke transported eastwards over the continent and enhanced in concentration on the
two following days. This plume structure is simulated by CAMS reanalysis for BBA, but only extending
southwards to 30°S. After 5 September, the CO distributions also depicts the plume originating from
South America, located over the Atlantic (at 25-35°S, **Figure S2c, d**). This plume presents similar CO
amounts as that of the river of smoke (3-3.5 $10^{18}$ molecules cm$^{-2}$) and it is transported eastwards until
reaching the western South African coast on 6 September (Chazette et al., 2019). This is qualitatively
consistent with the BBA plume shown by CAMS reanalysis, but these last ones show a relatively less
dense plume as compared to that of the river of smoke.
On 5 September, the space-borne lidar CATS overpassed southern Namibia at around 2200 UTC across
the mid-level cloud band (**Figure 7a**), just to the south of the Bonanza AERONET station (**Figure 6c**).
CATS provides further observational evidence that BBA dominate the aerosol composition of the low
troposphere over continental southern Africa during the period of interest (**Figure 7b**). Smoke is seen
to be well mixed over the depth of the boundary layer (~3 km) over northern Botswana and Zambia



where fires are observed to be very active and widespread (**Figure 6d**). Interestingly, the depth of the
smoke layer is seen to be much deeper over continental Namibia, its top reaching almost 7.5 km AMSL
in the vicinity of the mid-level cloud band. The smoke is observed to reach the coastline above 3 km
AMSL, consistently with MODIS observations. At lower altitude, CATS evidences the presence of
pollution at the coast and maritime aerosols further west of the ocean (**Figure 7b**).
The river of smoke is anchored over Angola and northern Namibia, the latter location being where
most airborne and ground-based observations were acquired in the course of the AEROCLO-sA
campaign. While the CATS observations suggest that further south the BBA layer is advected eastward,
it appears from the CAMS reanalysis and MODIS observations that the region where BBA feeds into
the river of smoke does not move significantly. This is confirmed by the fact that large AOTs are
observed over northern Namibia (AERONET sun-photometer station in Windpoort, **Figure 8a**) between
3 and 7 September, as opposed to the other AERONET stations further south (Bonanza, HESS,
Uptington, **Figure 8b,c,d**) where more sporadic AOT peaks are observed, suggesting a propagating BBA
feature over this area. Large BBA-related AOT values are seen with CAMS in the vicinity of the Etosha
pan region (**Figure 4c,d**) that correspond to the maximum in AOT seen with AERONET over the station
of Windpoort (**Figure 8a**). The aerosol load is seen to increase over Windpoort from 1 to 6 September
(when it reaches 1.75 at 500 nm) and to decrease thereafter. The AOT values in Windpoort are
systematically higher than AOTs derived from other AERONET stations further south. The timing of the
AOT peaks obtained with CAMS is consistent with its sun-photometer-derived counterpart. The AOT
peak in the CAMS reanalysis suggests that the smoke river overpassed the HESS site around 1200 UTC
on 5 September, the Upington site at 0000 UTC on 6 September and the Bonanza site at 1800 UTC on
the same day, suggesting an eastward drift of the smoke river. The data from the AERONET sites is
consistent with the MODIS data in both magnitude and timing of the increased AOT values (**Figure
6c,e**). The river of smoke is clearly observed as an identifiable isolated feature sweeping through
Upington, i.e. away from the fire emissions, as opposed to what is observed further north in Windpoort
which is closer to Angola and Zambia. The good agreement between the AOT from MODIS and CAMS
at Windpoort and Bonanza (**Figure 8a, b**) is expected as MODIS AOTs are assimilated in CAMS retrieval
algorithm. Good agreement is also found between CAMS and AERONET AOTs at Upington (**Figure 8d**),
and to a lesser extent at HESS and Bonanza (**Figure 8b,c**, especially before 6 September). Windpoort
(**Figure 8a**) is an exception with AERONET AOTs differing significantly at times from the MODIS and
CAMS retrievals. The higher sun-photometer AOTs are most likely to be BBA dominated, a fact that is
highlighted by the average Angstrom coefficient values derived from the AERONET observations in
Windpoort of 1.7 (between 440 and 870 nm) which is consistent with previous finding evidencing that
the absorption Angstrom exponent of biomass smoke is typically between 1.5 and 2 (e.g. Bergstrom et
al., 2007).
4.2 Airborne observations and model simulation
In the latter stage of the CoL intrusion over Namibia (5 and 6 September), a large scale mid-level cloud
band is visible along the southern Africa western coastline at 0900 UTC on 5 September as already
highlighted using MODIS (**Figure 6c**) and confirmed using SEVIRI (**Figure S3a**), i.e. around the time the
Falcon 20 aircraft flew on that day. The presence of this cloud band is related with the presence of the
TTT (Ratna et al., 2013) and is triggered by the arrival of an upper-level trough over southern Africa
associated with the band of divergence east of its leading edge. Nearly cloud-free conditions are
observed over northern continental Namibia at this time. The mid-level cloud band moved inland
rapidly, together with the river of smoke, and covered a large part of Namibia in the afternoon as
shown with SEVIRI at 1600 UTC (**Figure S3b**). It sweeps through Namibia overnight and is observed
over eastern Namibia at 0600UTC on 6 September (**Figure S3c**) before starting to disintegrate. Almost
cloud-free conditions are seen after 1200 UTC (the cloud band is no longer visible in **Figure S3d**), at the
time of the Falcon 20 aircraft flight on that day. On 5 September, airborne observations acquired in
the morning are representative of conditions ahead of the cloud band sweeping across Namibia, while



the airborne data acquired in the afternoon of 6 September were acquired over Namibia after its
passage.
The observations made with the airborne lidar LNG over Windpoort and the Etosha region on 5 and 6
September (**Figure 9a, c**, respectively) clearly evidence the complexity of the layering within the BBA-
laden air masses. On the morning of 5 September the BBA layer is observed between ~2 and 6 km
AMSL and to be separated from the surface (reaching an elevation of 1.5 km AMSL over the plateau)
by a shallow developing convective boundary layer in which dust emission from the Etosha pan are
observed (Formenti et al., 2019). On the other hand, the BBA layer is separated from the surface away
from the plateau and over the ocean (beginning and end of flight). On the afternoon of 6 September,
the BBA layer is clearly observed to be mixed all the way down to the surface over the plateau and to
reach ~6 km AMSL, thereby extending over a depth of nearly 4.5 km. The vertical structure of the
Meso-NH-derived black carbon tracer along the flight tracks on both days is given in **Figure 9b, d**. On
5 September, the greater vertical extent of the BBA layer over the continental plateau (with respect to
the surrounding lower lands and ocean) at the beginning and the end of the flight is well captured in
the model (**Figure 9b**). Likewise, the decrease of the BBA layer over the sloping terrain towards the
ocean at the end of the flight on 6 September is also well represented (**Figure 9d**). On 6 September,
large concentrations of black carbon tracers are mixed all the way to the surface, as in the
observations, in the northern part of the flight (i.e. overpassing the Etosha pan). In contrast, the larger
black carbon tracer concentrations do not reach the surface in the morning, which was also observed
with the airborne lidar.
4.3 Analysis of BBA transport on 5 September
Meso-NH-derived backward trajectories ending at 0900 UTC 5 September 2017 along the F06 flight
track and at altitude between 1 and 5 km AMSL (**Figure 10a, c, d**) show that over the previous 3 days
the air masses documented with the airborne lidar LNG during the flight had travelled over regions
with detectable active fires (Figure 5) and where black carbon emissions are high in the GFED4s
inventory (**Figure S4**). The trajectories are transport emissions from Angola, Zambia, Zimbabwe and
Mozambique where fires are identified with MODIS (**Figure 6**). Airborne observations and simulations
show unambiguously that the atmospheric composition below 5 km AMSL along the aircraft flight track
is dominated by BBA from Angola and Mozambique (see **Figure 6)** coming from altitudes below 5 km
AMSL and swirling anticyclonically around a high pressure in the lower troposphere (**Figure 10**). It is
worth noting that the air masses ending at 4-5 km AMSL along the aircraft track all experience upgliding
in the previous 24 h (**Figure S5a**) while the black carbon tracer concentration is gradually increasing
over the 3 days for all air masses below 5 km AMSL (**Figure S5b**). At higher altitudes (i.e. 6-7 km AMSL),
backward trajectories ending north of the Etosha pan also start from Angola and Zambia below 5 km
AMSL (**Figure 10b**) while those ending south of the Etosha pan originate from higher altitudes
(between 8 and 11 km AMSL) and from the southeast, having travelled over Botswana, Mozambique
and South Africa, and to swirl cyclonically around the location of the CoL while descending along its
poleward fringes (recall that the CoL is located between Henties Bay, Bonanza and Windpoort on 3-4
September (**Figure 3c, d**)). The downgliding experienced by the air masses occurs in the previous 2
days (**Figure S4b**). The ERA5 data show the presence of ascending motion to the north and east of the
CoL centre and descending motion to the south and west of the CoL centre (see Section 3).
Airborne lidar observations evidence that over the ocean, the height of the cumulus clouds is observed
to coincide with the top of the BBA layer (**Figure 9a**). The dropsonde-derived RH and potential
temperature profiles acquired over the ocean (**Figure 11a**) show the presence of a strong RH and
temperature inversion at 6 km AMSL, topped by extremely dry air layer from the west (270°),
consistent with the backward trajectories seen in the southern part of the F06 flight track (**Figure 10b**).
This suggests a descent of upper tropospheric air along the southern fringes of the CoL. Directional
wind shear is observed in the BBA layer with northerly winds at the bottom (~2 km AMSL) and north-
northwesterly winds near the top (~6 km AMSL, **Figure 11b**), with the BBA layer being advected
towards the ocean at the speed of ~15 m s$^{-1}$. Over the Etosha pan, dry upper tropospheric air is



observed above 7 km AMSL, with significant RH above 6 km AMSL (**Figure 11c**). This, together with the
lidar-derived ABC observations, suggests that the BBA layer top can reach to almost 7 km AMSL
(enhanced ABC is observed above 6 km AMSL, i.e. above isolated cumulus-type clouds forming over
land). The wind direction in the air mass encompassing the upper part of the BBA layer is seen to be
remarkable consistent with winds from north-northwest between 4 and 7 km AMSL (**Figure 11d**), in
agreement with the backward trajectories in the northern part of the F06 flight track (**Figure 10a**). In
conclusion, the dropsonde-derived RH profile over Etosha between 5.5 and 7 km AMSL suggests large
scale ascending motion above the BBA layer over the continental plateau, as opposed to the nearby
ocean where the RH profile suggests strong subsidence associated with the South Atlantic high.
Enhanced ABC above 6 km AMSL is not related to differential transport of BBA layers of different origin,
but rather to the lifting of the top of the BBA layer (with non-negligible RH values contributing increase
lidar backscatter signal by hydroscopic growth of aerosols). The presence of black carbon tracers above
6 km AMSL is also seen in the Meso-NH simulation (**Figure 9b**). Backward trajectories computed
between 6 and 7 km AMSL in that area (between kilometres 400 and 1000 in **Figure 9a**) are nearly all
associated with air masses from the northeast, i.e. the fires prone regions of Angola and Zambia. These
backward trajectories are seen to upglide to almost 8.5 km AMSL in the 24 h preceding their arrival
over the Etosha pan region (not shown). This provides further confirmation of ascending motion above
the BBA layer over the Etosha pan region, consistent with air moving in the easterly low wave.
4.4 Analysis of BBA transport on 6 September
**Figure 12** shows the backward trajectories ending at 1200 UTC between 1 and 7 km AMSL, along the
F09 flight track on 6 September. Backward trajectories between 1 and 5 km AMSL (**Figure 12a, c, d**)
are very similar to those seen on 5 September, with air masses having travelled over Angola, Zambia,
Zimbabwe and Mozambique before reaching northern Namibia. The black carbon tracer
concentrations increase along the path of the trajectories towards Namibia (**Figure S5d**). On both 5
and 6 September, the air masses originate from the north east (**Figure 12**), ending between 4 and 5
km AMSL. All trajectories show significant lofting in the previous 24 h prior to reaching the area of the
Falcon flight. Above, the backward trajectories ending along the flight track are essential coming from
the southeast and are descending from 11 to 7 km upon reaching the area of interest (**Figure 12b**).
Only a few trajectories ending in the Etosha pan region originate from the northeast (as opposed to
the previous day when a significant number of such trajectories were seen). On this day, vertical
motion above the BBA layer is dominated by subsiding dry air masses travelling along the southern
fringes of the CoL during the previous 3 days (**Figure S5c**). On 6 September, these descending
trajectories extend much further over the ocean than on the previous day (compare **Figure 10b** and
**Figure 12b**). This is consistent with the fact that the CoL was well established over the area of the flight
two days prior to the flight (i.e. on 4 September, **Figure 3d**), more so than the CoL on 3 September
(**Figure 3c**) two days prior to the flight on 5 September. As a result of the dominance of the descending
air masses over northern Namibia, a sharp RH transition to very dry conditions is observed at the top
the BBA layer, above 6.5 km AMSL, along the southern part of the F09 flight track (**Figure 11e**).
Likewise, the top of the BBA layer over Etosha is significantly lower than on the previous day, even
though the structure of the RH profile in the upper part of the BBA layer suggest weaker ascending
motion (**Figure 11f**) than on 5 September. The RH and potential temperature profiles acquired over
the Etosha pan confirm the presence of a deeper convective boundary layer on 6 September (with a
top at 4 km AMSL) due to the fact that the flight took place later in the day than on 5 September (the
top of the convective boundary layer being observed at 2 km AMSL). The maxima of RH are observed
to be slightly higher on 6 September (~80%, **Figure 11f**) than on the previous day (~70%, **Figure 11c**).

**5. The impact of the CoL and the TTT on the tropospheric composition over the sub-continent**
In the previous section, we showed that the mid-tropospheric circulation associated with the presence
of the CoL potentially modulates the depth of the underlying widespread layer of the smoke layer over


northern Namibia. The modulation appears to be forced by the vertical motion associated with the
CoL, i.e. subsiding air parcels to the south and west of its centre, and ascending motion to the north
and east of it. In the following, we provide further evidence of this modulation over northern Namibia,
where the CoL was observed to be intense during two days (3 and 4 September) by looking at the
evolution of the vertical distribution of black carbon tracers and potential vorticity, among other
variables, using Meso-NH simulations and ERA5 reanalysis.
The horizontal extent of the CoL over Northern Namibia and more particularly above three sun-
photometer stations (namely Henties Bay, Windpoort and Bonanza) on 3 and 4 September 2017 was
highlighted using ERA5-derived potential vorticity (PV) in **Figure 3**. The time-height section of the PV
over Windpoort from 1 to 6 September (**Figure 13a**) evidences the presence of the CoL from mid-day
on 2 September until the end of 4 September, associated with negative PV (less than $4\ 10^{-6}$ K m$^2$ kg$^{-1}$ s$^{-1}$
) and cyclonic circulation, between 400 and 300 hPa. In the lower troposphere, the shallow nocturnal
boundary layer is also associated with negative PV and cyclonic circulation. The top of the nocturnal
boundary layer is found around 875 hPa. In comparison, the deeper daytime PBL is associated with
positive PV and anticyclonic circulation. The strongest ascending motions in the mid-troposphere (800-
400 hPa) is seen between 1200 UTC on 2 September and 1200 UTC on 3 September (**Figure 13b**). This
strong vertical motion is associated with the sloping negative PV structure related to the incoming CoL
(**Figure 13a**). Concomitantly, RH above 600 hPa increases dramatically up to 400 hPa at 1200 UTC on 3
September (**Figure 13c**), together with the cloud cover (**Figure 13d**). The cloud cover is found to
decrease after 3 September, unlike RH that remains high above 600 hPa in the following days. The
peak in cloud cover over Windpoort on 3 September is consistent with the space-borne observations
made with SEVIRI (**Figure S6b**) which shows the presence of an isolated patch of mid-level clouds south
of the Etosha pan region on 2 September at 1200 UTC and over the area of Etosha on 3 September at
1200 UTC (**Figure S6a,b**). Overall, the mid-level cloud cover above continental Namibia is low and the
isolated nature of the mid-level cloud patches on 2 and 3 September suggests that they are formed
locally rather than being advected from another area. On 4 September, the space-borne lidar CATS
overpassed northern Namibia, just to the north of the Etosha pan during the daytime and across the
CoL (**Figure 6a**). The CATS observations evidence that the PBL is the deepest seen over the continental
plateau along the transect (**Figure S6d**) and that mid-level clouds are forming on top of the PBL to the
north of the centre of the CoL. This is also a strong evidence that the CoL generates strong local
ascending motion in the lower troposphere leading to the formation of an isolated cloud patch, also
visible on the SEVIRI images (**Figure S6c**).
The Meso-NH-derived time-height evolution of smoke tracer concentrations, vorticity and cloud liquid
water over Windpoort between 1 and 6 September 2017 is shown in **Figure 14a**. Even though the
largest black carbon tracer concentrations are seen after 1200 UTC on 5 September (i.e. the time when
the airborne observations discussed above were acquired), the simulation shows that the deepest BBA
layer over Windpoort occurred late on 3 September (1200-2100 UTC) as well as late on 4 September
(1800-2100 UTC) in connection with the presence of CoL-related potential vorticity in the upper-
troposphere reaching towards the surface to altitudes between 6 and 7 km AMSL. On this occasion,
the top of the BBA layer reaches at least 8 km AMSL, while later on, i.e. on 5 and 6 September, the top
of the BBA layer is between 6 and 6.5 km AMSL as with the airborne lidar measurements in the area
of Windpoort. The liquid water content at the top of the BBA layer increases from 1 to 3 September,
together with the height of the condensation level (**Figure 14a**), both variables reaching their
maximum values late on 3 September, i.e. shortly after the descent of mid-tropospheric vorticity on
that day. On 4 and 5 September, the level of condensation is much lower (between 4 and 6 km AMSL)
and within the BBA layer, in accordance with the lidar observations. Further south, in Upington (**Figure
14b**), the presence of mid-tropospheric potential vorticity at altitudes as low as 7 km AMSL around
mid-day on 5 September. Later that day, Meso-NH simulates the deepest BBA layer of the period, the
top of the layer reaching between 7 and 8 km AMSL. The simulation reproduces consistently the
sporadic nature of the event in Upington associated with the advection of the river of smoke observed
in Section 4, with large black carbon tracer concentrations occurring late on 5 September, over a large





depth in the lower troposphere. It is worth noting that liquid water is seen to extend from 4 to 7 km
AMSL, thereby suggesting thick clouds embedded in the river of smoke, in accordance with
observations from MODIS (horizontal distribution, **Figure 6**) and from CATS (vertical distribution,
**Figure 7**) as well as from SEVIRI regarding clouds (**Figure S6**).
The characteristics of the river of smoke (timing, vertical extent of the BBA layer) seen in Upington
based on Meso-NH simulations cannot be seen further north (e.g. in Windpoort) as northern Namibia
was under the influence of a well formed, stationary, isolated CoL on 3 and 4 September (**Figure 3c,**
**d**), unlike South Africa over which the fast evolving CoL travelled south-eastward between 5 and 6
September while merging back with the main westerly flow (**Figure 3e, f**). Hence, the picture emerges
that the characteristics of the river of smoke are very much tied to the later (fast evolving) stage of the
evolution of the CoL than the earlier (stationary) stage. The model results discussed in this section
highlight the mechanisms by which the CoL observed over southern Africa at the beginning of
September 2017 influences the vertical structure of the BBA layer, essentially through the CoL-related
ascending/descending motion above the BBA layer. The deepest BBA layers over northern Namibia
could not be observed with the airborne platform operated during AEROCLO-sA. Nevertheless, the
Meso-NH simulation for the period 5-6 September being very consistent with the observations
gathered during the campaign, we are confident that the Meso-NH-derived structure of the BBA layer
over northern Namibia during the stationary phase of the CoL is realistic.

**6. Conclusions**
The formation of a river of smoke crossing southern Africa has been investigated during the Aerosols,
Radiation and Clouds in southern Africa (AEROCLO-sA) campaign for 2 to 6 September 2017 in
connection with a mid-level TTT and a CoL, using a complementary set of global and mesoscale
numerical simulations as well as ground-based, airborne and space-borne observations.
Numerical simulations performed with the high resolution Meso-NH model together with space-borne
lidar observation made with CATS provide evidence that the top of the BBA layer over northern
Namibia (where the CoL remained stationary for 2 days) may reach altitudes higher than of 8 km AMSL.
This is much higher that the height of the top of the BBA layer over the regions where the smoke
originates from (Angola, Zambia, Zimbabwe, Mozambique), i.e. ~5 to 6 km AMSL. The impact of the
CoL-driven TTT on the vertical distribution of BBA was also felt further south, over South Africa, in the
form of a river of smoke as the CoL was rapidly travelling southwestward before merging with the
westerlies. The TTT created favourable conditions for efficiently transporting BBA-prone tropical air
masses towards the southwestern Indian Ocean. The temporal evolution of the river of smoke was
found to be connected to the fast evolving stage of the CoL. Besides favouring the increase of the BBA
layer top, the ascending motion associated with the CoL also promoted the occurrence of mid-level
clouds over northern Namibia in the early (stationary) phase of the CoL evolution (2-4 September,
**Figure 15a**) while cloud-free conditions were observed everywhere else over the continent. In the fast
evolving stage of the CoL (5 and 6 September, **Figure 15b and 15c**, respectively), a band of mid-level
clouds was embedded in the river of smoke that were related to the circulation in the lee of the CoL.
Even though the CoL observed on 3-6 September 2017 did not impact the circulation at the surface
directly, the Meso-NH simulation provides unambiguous evidence that the river of smoke event that
swept through western South Africa (i.e. away for the region of fires) not only had a mid-tropospheric
signature, but also that air quality associated with the transported BBA was reaching the surface, as
illustrated in Upington. Such behaviour was also observed during SAFARI 2000 by Magi et al. (2003)
and Schmid et al. (2003). These authors have also shown that the height of the top of the BBA layer
decreased significantly from northwest to southeast, i.e. between fire-prone regions in the Tropics and
the exit region of the river of smoke. The altitude of the top of the BBA layer observed during AEROCLO-
sA is consistent with that measured over Zambia during SAFARI2000 (Schmid et al., 2003).



To the author's knowledge, this is the first time the link between the CoL dynamics and the formation
of a river of smoke is established. The CoL was an essential ingredient of the TTT that developed across
southern Africa. TTTs are known to compose the dominant rainfall-producing weather system over
southern Africa during the austral summer. Here, we demonstrate that TTTs also play a role in the
transport of BBA during the winter.
Future research will aim at consolidating our understanding of the impact of the main dynamical
features highlight in this study (CoL, TTT, Angola low) on the formation and the evolution of rivers of
smoke over the southern Africa sub-continent. Among overarching open questions to be investigated
using ERA5/CAMS reanalysis and satellite aerosols products, we shall assess: (1) what is the frequency
"rivers of smoke" during the austral winter, (2) how important is this mechanism for the transport of
CO and aerosols out of southern Africa compared to the other transport patterns identified by
Garstang et al. (1996), (3) whether CoLs and/or TTTs are systematically associated with rivers of smoke
and (4) what is the importance/role of the Angola low in promoting the accumulation of BBA in the
tropical band prior to their injection in smoke rivers.



**Data availability.** The aircraft data acquired specifically in the framework of the project and used here
can be accessed via the AEROCLO-sA database at https://baobab.sedoo.fr/AEROCLO/ and now have
DOIs. The LNG lidar data DOI is https://doi.org/10.6096/AEROCLO.1774. The dropsondes data DOI is
https://doi.org/10.6096/AEROCLO.1777. The Meso-NH-derived fields and back trajectories data can
be obtained upon request to the corresponding author of the paper. AERONET products can be
accessed at https://aeronet.gsfc.nasa.gov/. IASI data can be obtained via the AERIS data center
(https://iasi.aeris-data.fr). SEVIRI imagery can be accessed via http://aeroclo.sedoo.fr/. MODIS data is
accessible via https://giovanni.gsfc.nasa.gov/giovanni/. SEVIRI images are available via EUMETSAT
(European Organisation for the Exploitation of Meteorological Satellites).
**Author contributions.** CF processed and analysed the airborne lidar data and the dropsonde data as
well as the Meso-NH simulations, ERA5 and CAMS reanalysis, and wrote the paper. MG assembled the
material from ECMWF (ERA5 and CAMS), prepared the related figures, and contributed to the
interpretation of the atmospheric dynamic and composition data. PC gathered the CATS lidar data and
the MODIS data, and produced the related figures. JPC performed the Meso-NH simulation and
produced the related figures. SJP contributed to the analysis of the synoptic conditions. JC collected
CO data from IASI and produced the corresponding figure. PF coordinated the AEROCLO-sA project. All
have contributed to the writing of the paper.
**Competing interests.** Paola Formenti is guest editor for the ACP Special Issue "New observations and
related modelling studies of the aerosol–cloud–climate system in the Southeast Atlantic and southern
Africa regions". The remaining authors declare that they have no conflicts of interests.
**Special issue statement.** This article is part of the special issue "New observations and related
modelling studies of the aerosol–cloud–climate system in the Southeast Atlantic and southern Africa
regions (ACP/AMT inter-journal SI)". It is not associated with a conference.
**Acknowledgments.** The authors thank the AERIS data centre for their support during the campaign
and managing the AEROCLO-sA database. Airborne data was obtained using the aircraft managed by
SAFIRE, the French facility for airborne research, an infrastructure of the French National Centre for
Scientific Research (CNRS), Météo-France and the French space agency CNES. The authors thank F.
Blouzon and A. Abchiche (DT-INSU) as well as P. Genau and M. van Haecke (LATMOS) for their support
in operating and processing the LNG data. The invaluable diplomatic assistance of the French Embassy
in Namibia, the administrative support of the Service Partnership and Valorisation of the Regional
Delegation of the Paris-Villejuif Region of the CNRS, and the cooperation of the Namibian National
Commission on Research, Science and Technology (NCRST) are sincerely acknowledged. The authors
acknowledge the MODIS science, processing and data support teams for producing and providing
MODIS data (at https://modis.gsfc.nasa.gov/data/dataprod/). The authors thank the AERONET
network for sun-photometer products. IASI is a joint mission of EUMETSAT and the Centre National
d'Etudes Spatiales (CNES, France). The authors acknowledge the AERIS data centre for providing access
to the CO IASI data in this study as well as the Université Libre de Bruxelles and LATMOS for the
development of the retrieval algorithms.
**Financial support.** This work was supported by the French National Research Agency under grant
agreement n° ANR-15-CE01-0014-01, the French national program LEFE/INSU, the Programme
national de Télédétection Spatiale (PNTS, http://programmes.insu.cnrs.fr/pnts/), grant n° PNTS-2016-
14, the French National Agency for Space Studies (CNES), and the South African National Research
Foundation (NRF) under grant UID 105958. The research leading to these results has received funding
from the European Union's 7th Framework Programme (FP7/2014-2018) under EUFAR2 contract
n°312609. Computer resources for running Meso-NH were allocated by GENCI through project 90569.



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





**Table 1.** Correlation coefficients computed between the first and second principal components of
MSLP, geopotential height at 300 and 700 hPa and the first principal component of the organic matter
AOT. P-values are indicated in brackets.

|  | Explained variance | Correlation |
|---|---|---|
| Z300 | 56% | 0.50 (p<0.01) |
| Z700 | 74% | 0.78 (p<0.01) |
| MSLP | 43% | 0.01 (p=0.60) |






**Figures**

(a)                                              (b)

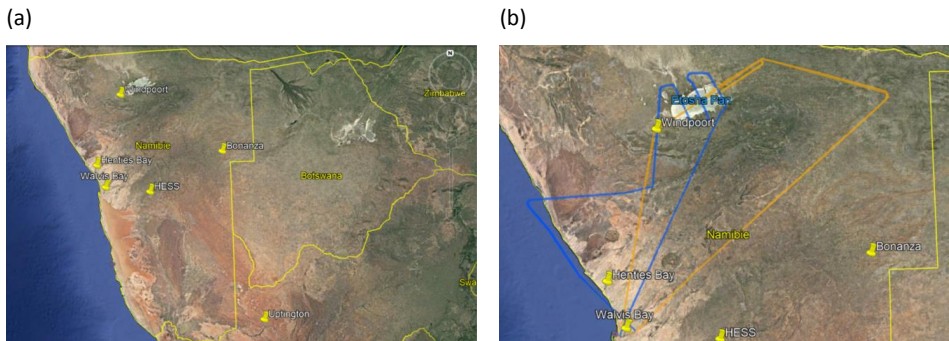

**Figure 1:** (a) Geographical map of Namibia and surrounding countries with the location of the main
sites of interest: Walvis Bay (airport), Henties Bay (AEROCLO-sA main ground-based supersite) and
AERONET stations in Windpoort, HESS, Bonanza (Namibia) and Upington (South Africa). (b) Zoom on
northern Namibia where the Etosha pan is located (white area just northeast of Windpoort). The
blue solid line represents the SAFIRE Falcon 20 flight track on 5 September in the morning (0736-
1014 UTC) and the orange solid line represents the Falcon 20 flight track on 6 September afternoon
(1055-1401 UTC). Map credit: © Google Earth 2021.




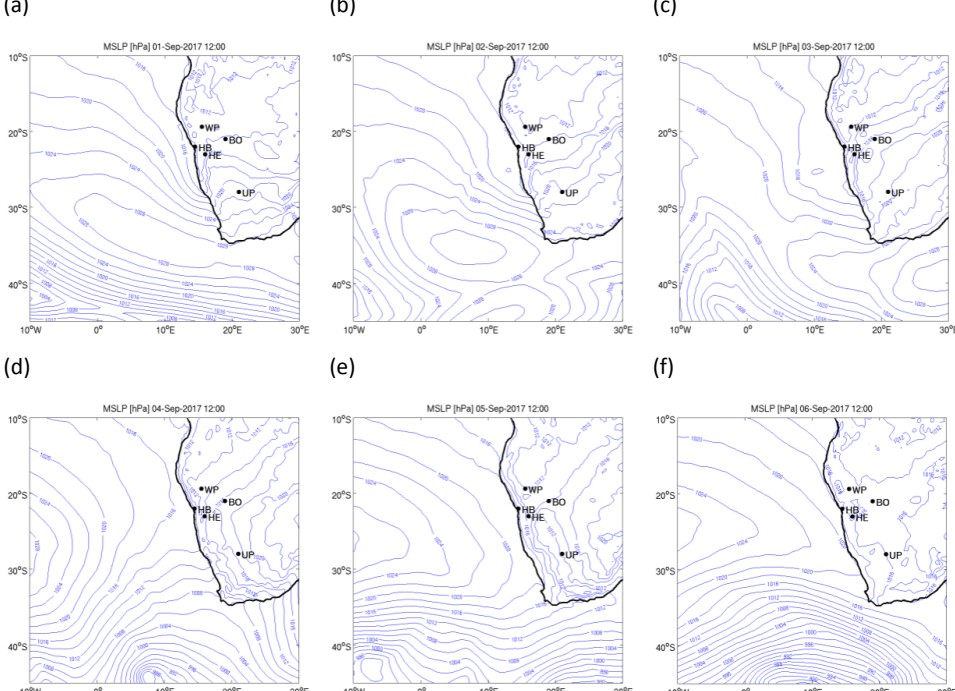

**Figure 2**: Mean sea level pressure (hPa) at 1200 UTC on (a) 1 September, (b) 2 September, (c) 3 September, (d) 4 September, (e) 5 September and (f) 6 September 2017, from ERA5 reanalysis. The names of the instrumented sites appear in black (from north to south): WP is Windpoort, BO is Bonanza, HB is Henties Bay, HE is HESS (Namibia) and UP is Upington (South Africa).




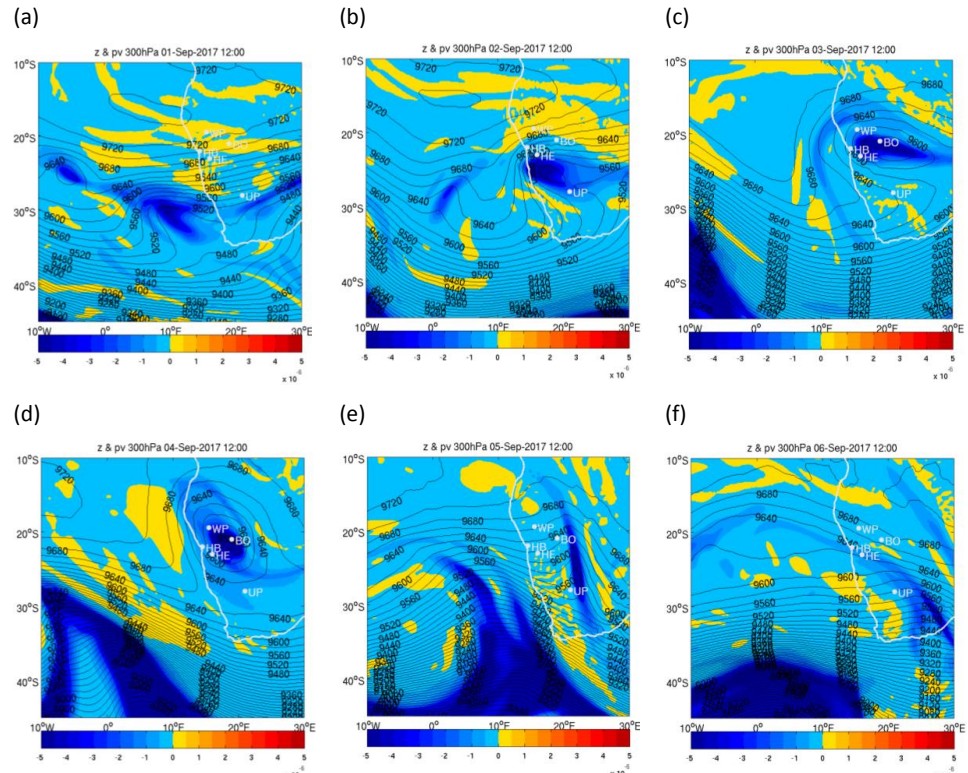

**Figure 3**: Geopotential (m, contours) and potential vorticity (K m$^2$ kg$^{-1}$ s$^{-1}$, colour) at 300 hPa at 1200 UTC on (a) 01/09, (b) 02/09, (c) 03/09, (d) 04/09, (e) 05/09 and (f) 06/09 2017, from ERA 5 reanalysis. The names of the instrumented sites appear in white (from north to south): WP is Windpoort, BO is Bonanza, HB is Henties Bay, HE is HESS (Namibia) and UP is Upington (South Africa).






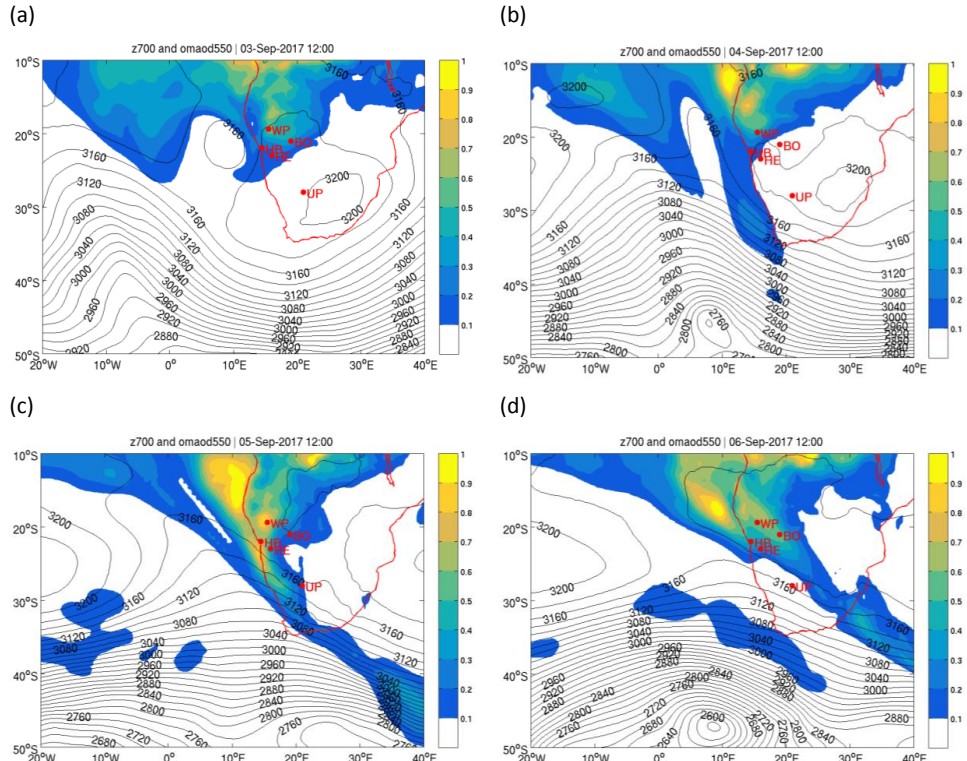

**Figure 4**: Geopotential at 700 hPa (contours) and organic matter AOT (colour) and 1200 UTC on (a) 03/09, (b) 04/09, (c) 05/09 and (d) 06/09 2017, from CAMS reanalysis. The names of the instrumented sites appear in red (from north to south): WP is Windpoort, BO is Bonanza, HB is Henties Bay, HE is HESS (Namibia) and UP is Upington (South Africa).






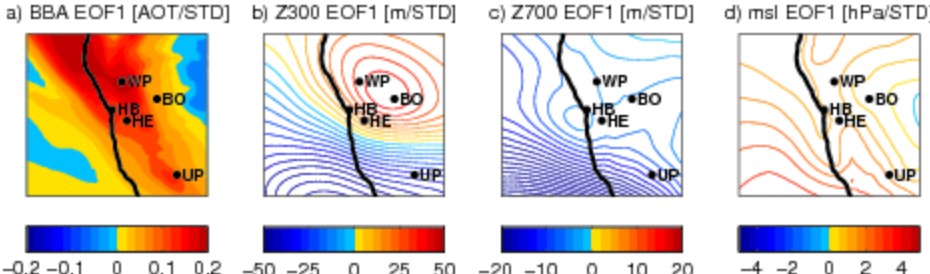

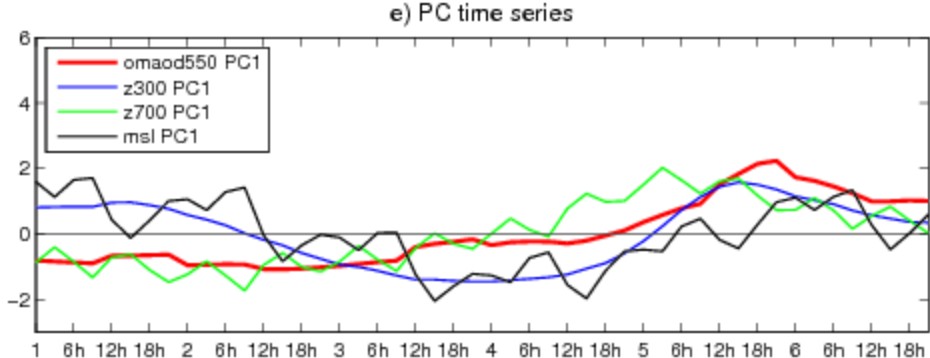

**Figure 5:** Principal component analysis (PCA) of CAMS BBA and atmospheric circulation above Namibia from 1 to 6 Sep 2017. In top panels, the anomaly patterns associated with the first mode of variability of (a) organic matter AOT at 550 nm, geopotential height at (b) 300 hPa and (c) 700 hPa, and (d) MSLP are obtained by regressing raw data onto the PCA time series displayed in panel (e). Anomaly patterns display anomalies per standard deviation (STD). The 12-h oscillation seen in the time series of the MSLP and the geopotential at 700 hPa is a tidal effect, typical of the Earth's atmosphere (Chapman and Lindzen, 1970).


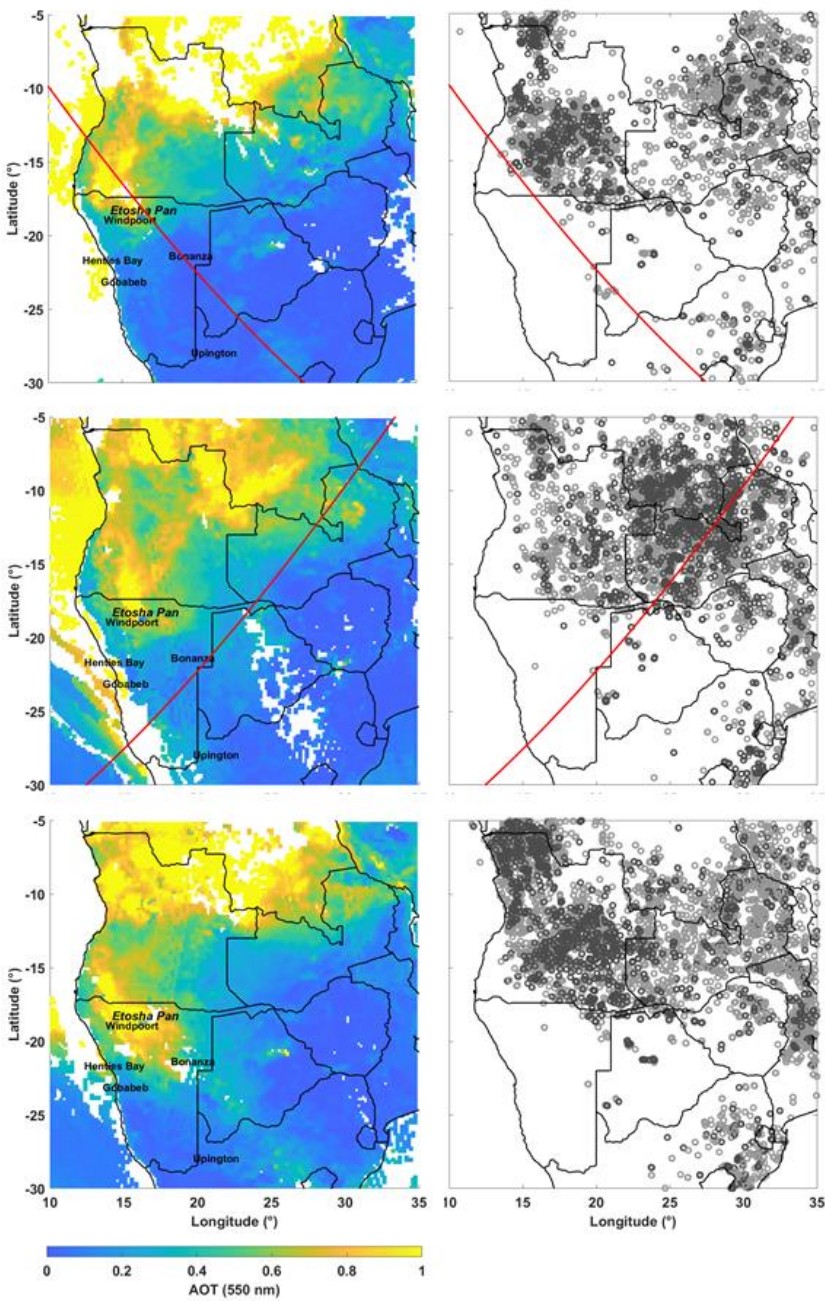

**Figure 6** MODIS-derived AOT (left panels) on and Fire hot-spots location (right panels) derived from MODIS on 4 (a,b), 5 (c,d) and 6 (e,f) September 2017. The CATS tracks overpassing Namibia on 4 and 5 September are overlain on (a,b) and (c,d) as a solid red line. The confidence in the detection of the location of the fire hot spots is indicated by the colour of the circles (dark circles indicating high confidence and grey circles, nominal confidence, as provided by the MODIS team).

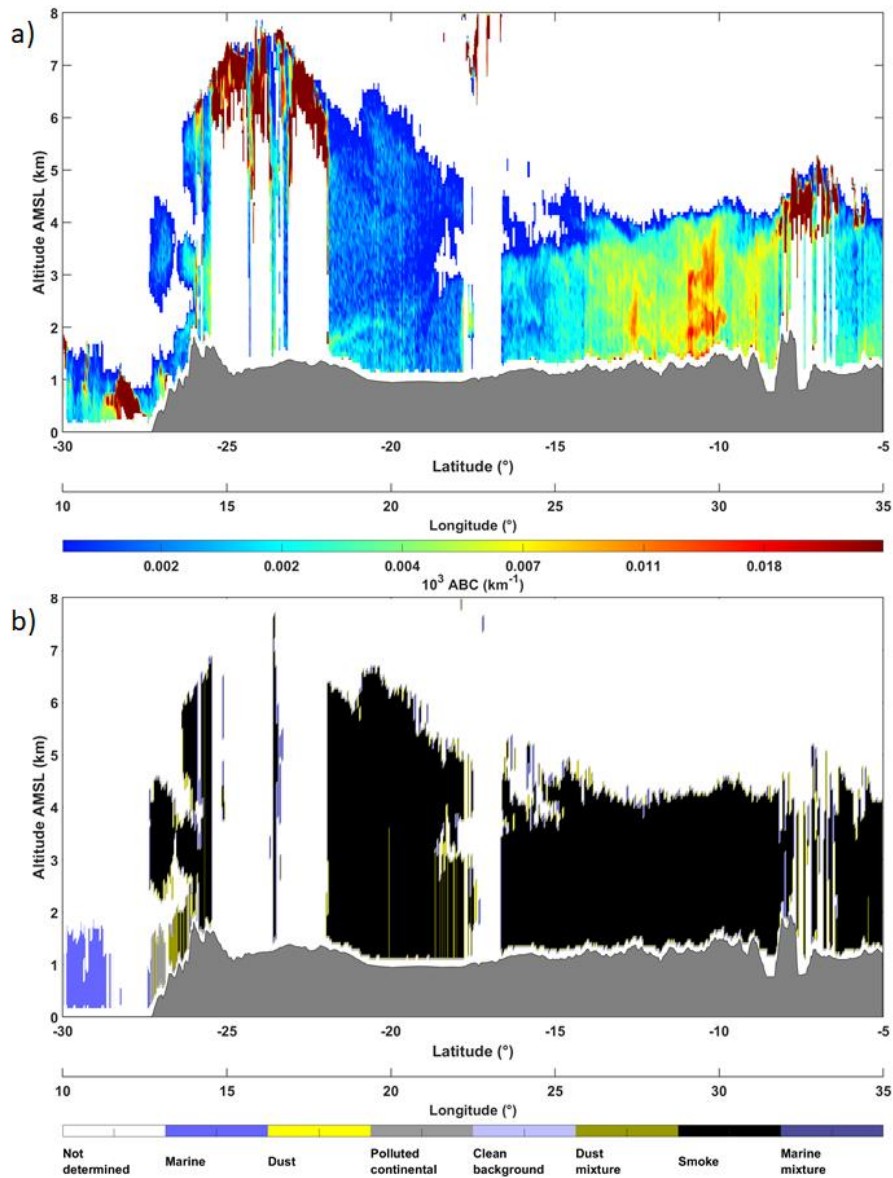

**Figure 7:** (a) Total attenuated backscatter coefficient from the space-borne lidar CATS between 22:05 and 22:21 UTC on 5 September 2017. (b) Same as (a), but for CATS-derived aerosol. The corresponding CATS track is shown in Figure 6b.






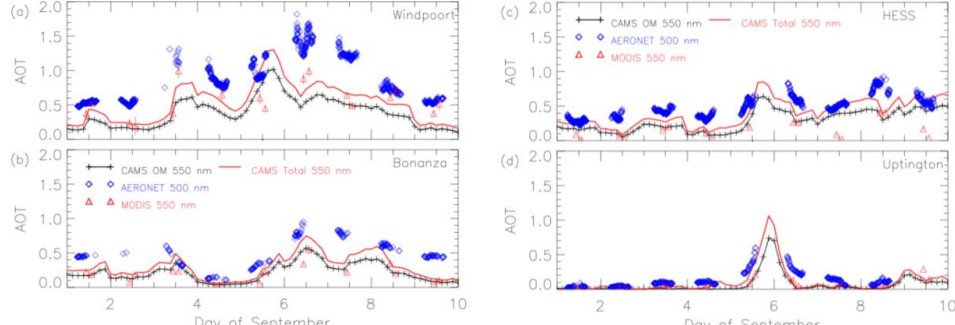

**Figure 8**: Time evolution of AERONET sun-photometer-derived total AOT (blue diamonds), MODIS AOT from Terra and Aqua (red triangles) and CAMS-derived total AOT (red solid line) and organic matter AOT (solid black line with black crosses) over (a) Windpoort, (b) Bonanza, (c) the HESS site and (d) Upington from 1 to 10 September 2017.



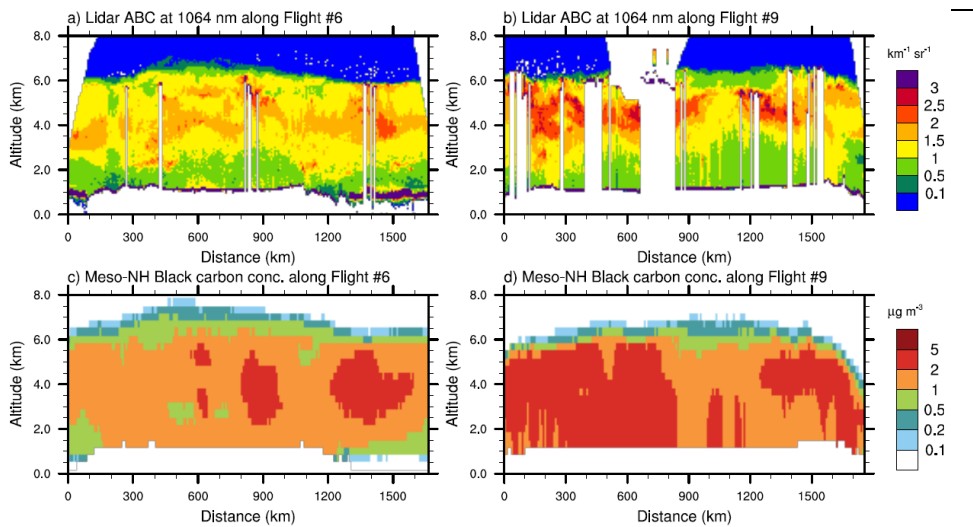

**Figure 9**: (a) Distance-height cross-section of attenuated backscatter coefficient derived at 1064 nm from LNG along the flight track of the Falcon 20 on 5 September from 0736 to 1014 UTC (see **Figure 1b**). (b) Black carbon tracer concentration as simulated with Meso-NH at 0900 UTC 5 September 2017 along the Falcon 20 flight track. (c) Same as (a) but on 6 September from 1055 to 1401 UTC. (d) Same as (b) but at 1200 UTC on 6 September.




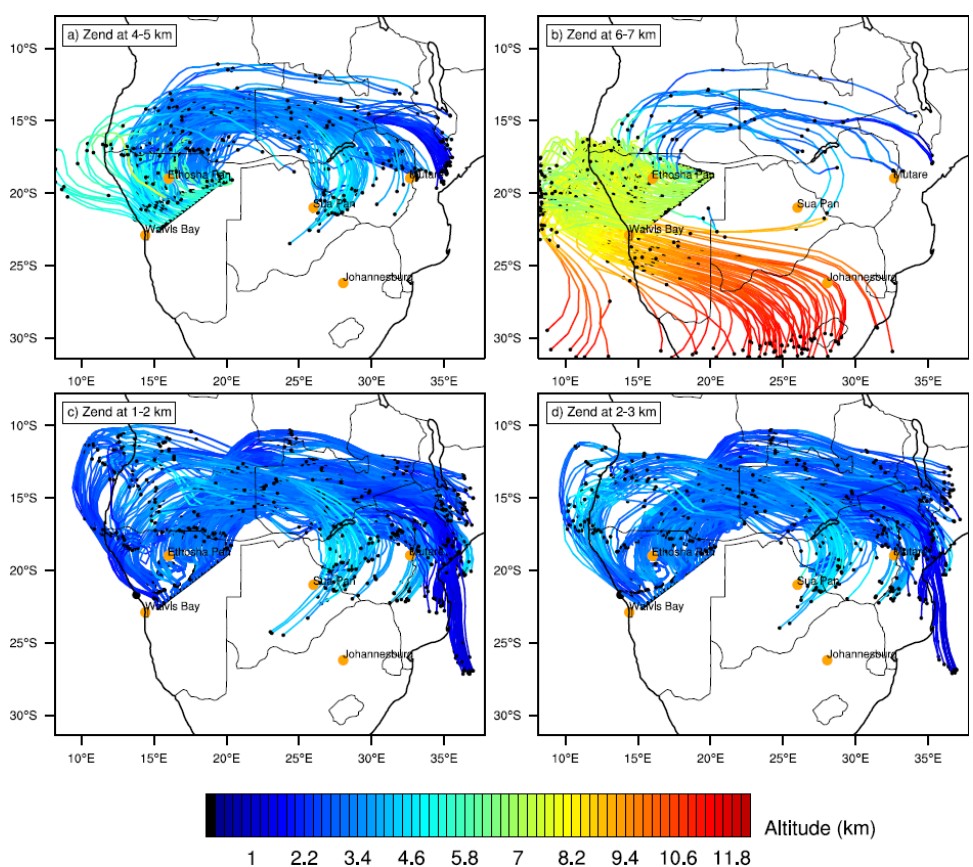

**Figure 10**: 102-h backward trajectories ending at 0900 UTC 5 September 2017 along the F06 flight track and at altitude between (a) 4 and 5 km AMSL, (b) 6 and 7 km AMSL, (c) 1 and 2 km AMSL and (d) 2 and 3 km AMSL. Dots on the backward trajectories are spaced at 24-h intervals. One backward trajectory out of 20 is plotted.






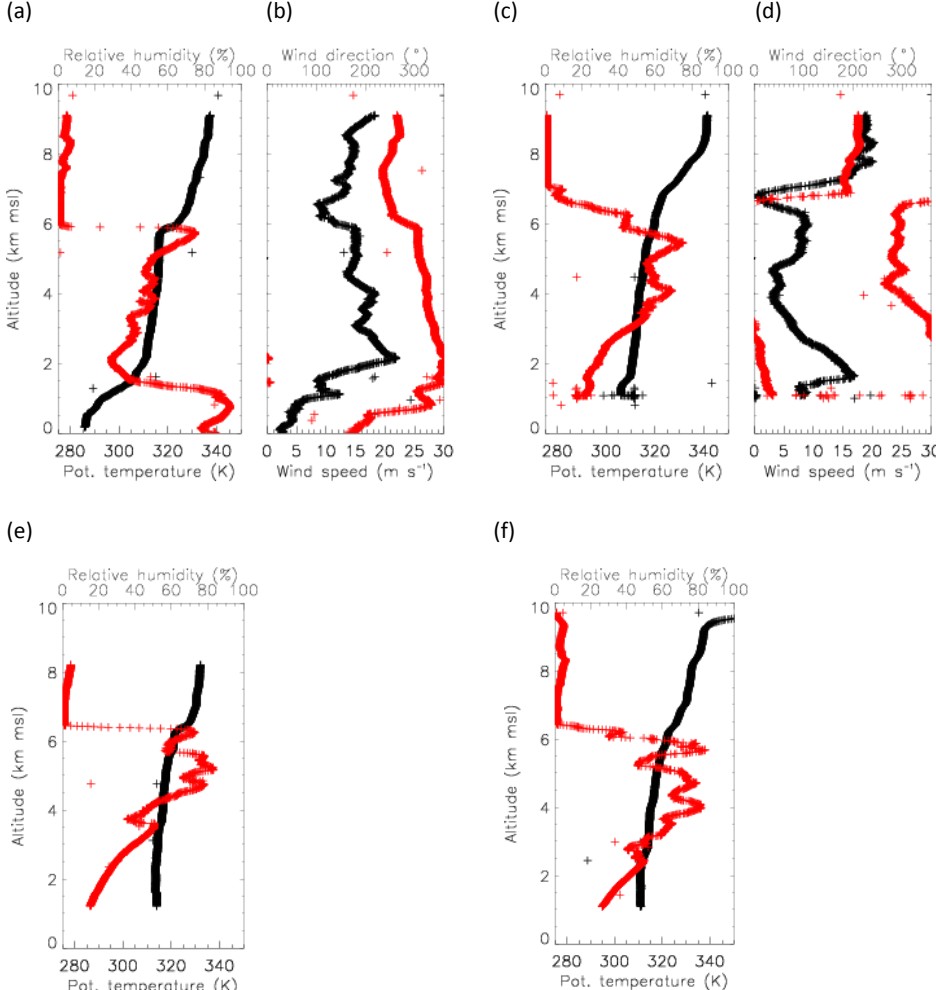

**Figure 11**: (a) Potential temperature (black) and relative humidity (red) profiles and (b) Wind speed (black) and wind direction (red) profiles derived from the dropsonde launched at 0951 UTC on 5 September. (c) and (d) same as (a) and (c), respectively, but for the dropsonde released at 0839 UTC. (e) Same as (a), but for the dropsonde released at 1337 UTC on 6 September. (f) Same as (e), but for the dropsonde released at 1146 UTC.


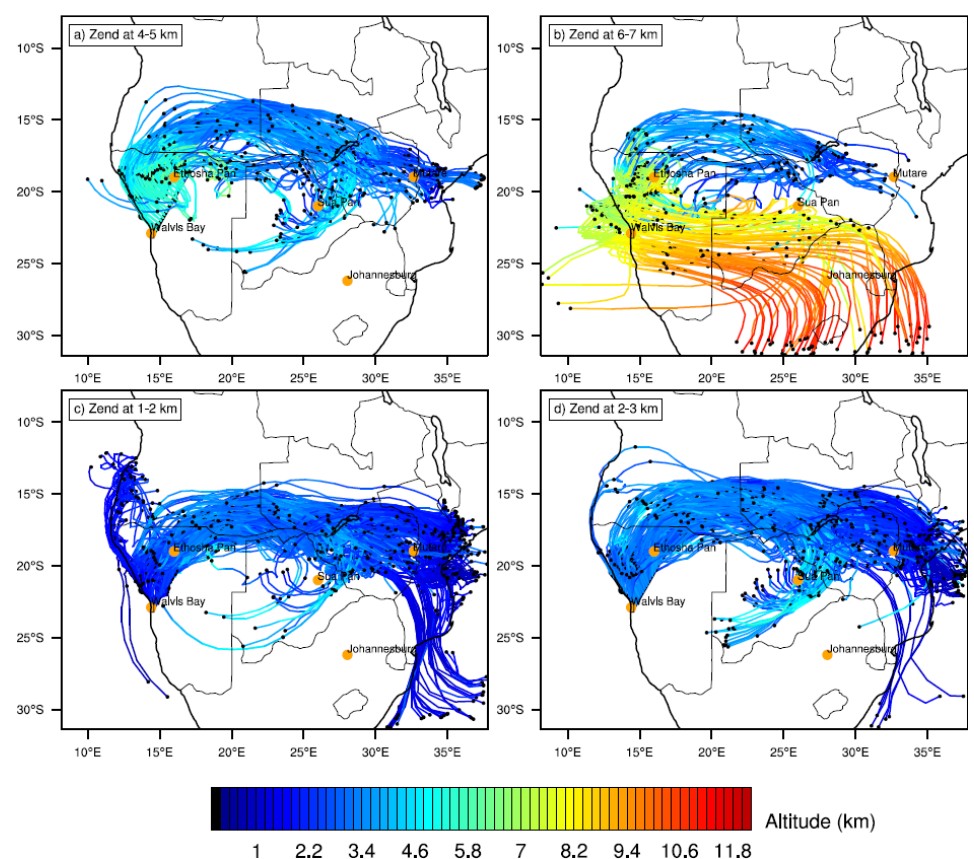

**Figure 12**: As in Figure 10, but ending at 1200 UTC 6 September 2017 for the F09 flight track.


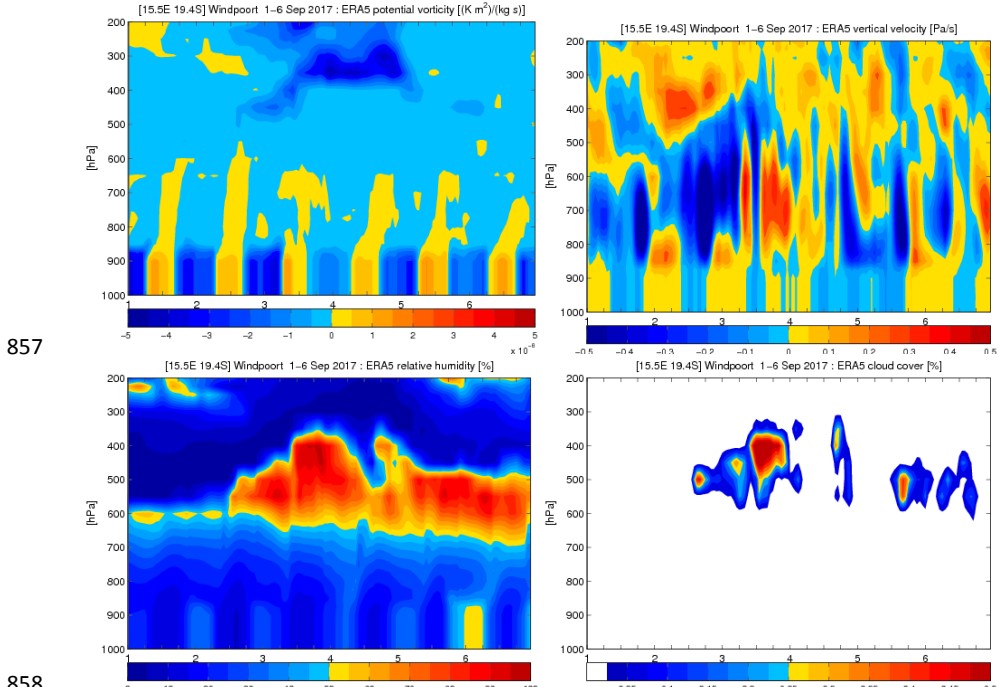

**Figure 13.** Time-height cross-section of (a) potential vorticity (K m$^2$ kg$^{-1}$ s$^{-1}$), (b) vertical velocity (Pa s$^{-1}$), (c) relative humidity (%) and (d) cloud cover over Windpoort between 1 and 6 September 2017 from hourly ERA 5 reanalysis. Ascending motions are associated with negative ω values.

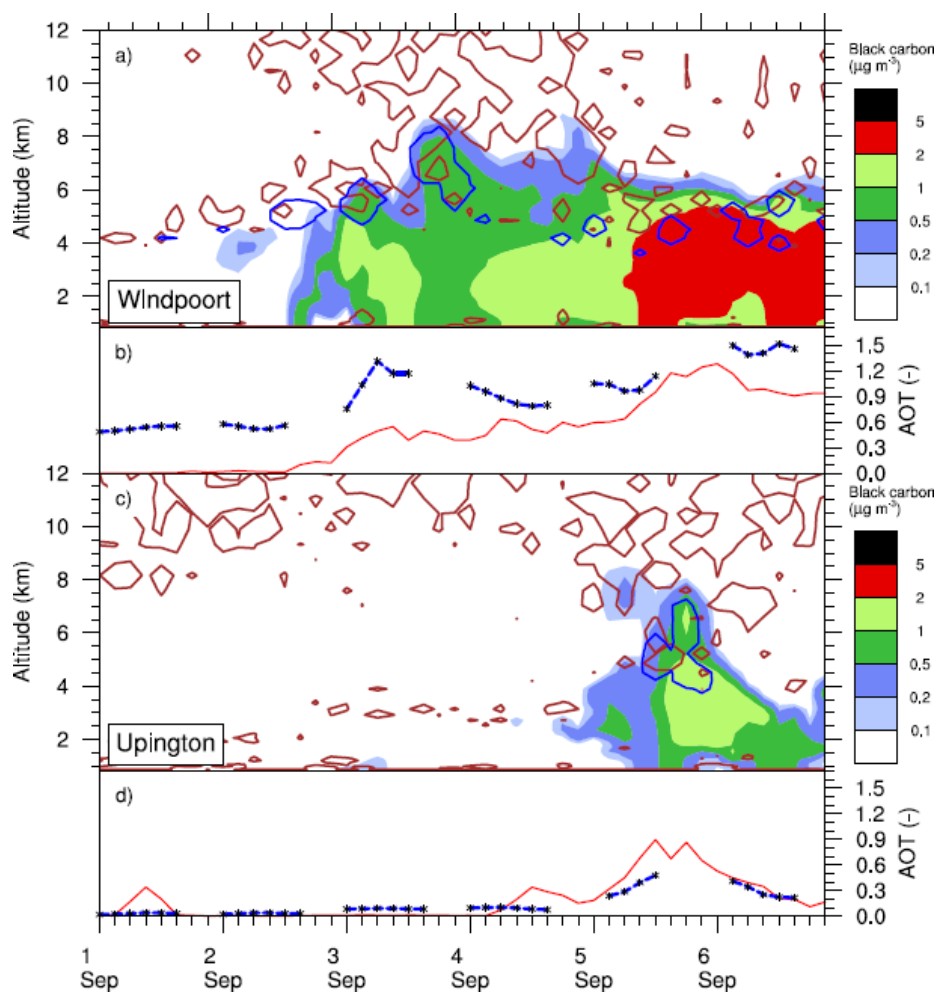

**Figure 14**: (a) Top panel: time height evolution of black carbon tracer concentration (colour) between 1 and 6 September 2017 from the Meso-NH simulation over Windpoort. Blue contours represent liquid water while brown contours represent potential vorticity. Bottom panel: AOT derived from Meso-NH (solid line, BBA only) and sun-photometer (blue crosses). (b) Same as (a), but for Uptington.

862




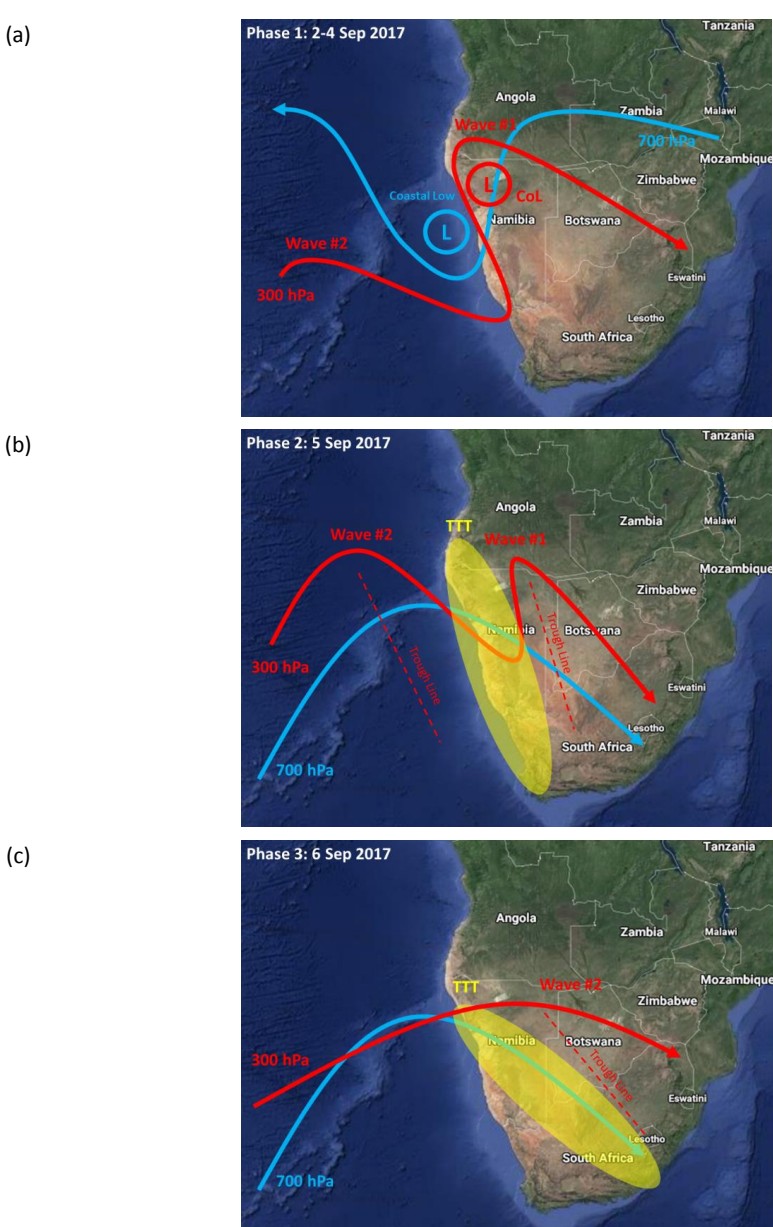

**Figure 15**: Map of southern Africa with the main dynamical features described in the text during the period 2-6 September 2017. (a) Phase 1 (2-4 September) with the easterly flow at 700 hPa (blue line with arrow) and the imbedded coastal low (blue L), together with the westerly waves at 300 hPa (red line with arrow) and the imbedded CoL (red L). (b) Phase 2 (5 September) with the location of the 2 westerly waves and associated trough lines (red dashed lines), the westerly mid-level flow at 700 hPa (blue line with arrow) and the location of the formed TTT (yellow shaded area). (c) Phase 3 (6 September) with the passage of the 2nd westerly wave and the associated trough line over the continent and the eastward displacement of the TTT. Map credit: © Google Earth 2021.

863