# Peer review of "Smoke in the river: an AEROCLO-sA case study"

_Atmospheric Chemistry and Physics, 2021_

## Author Comment (AC1)

**ACP-2021-939: Reply to Reviewer 1**

**Smoke in the river: an AEROCLO-sA case study, Cyrille Flamant et al.**

The authors would like to thank the reviewer for her/his valuable comments which helped improving the quality of the manuscript. Our point-by-point responses to the reviewer's comments appear in blue below. The text modified in the revised version of the MS and included in the response appears in red.

**General comments:**

The authors conducted a multifaceted study of an atmospheric long-distance mass (e.g., gas, aerosol) transport feature referred to as "River of Smoke" over southern African sub-continent. River of smoke refers to coherent bands of smoke (from biomass burnings) spanning hundreds of kilometers in width and extending for a few thousands of kilometers. Authors use a set of space-borne, aircraft-based, and ground-based observations in conjunction with global and mesoscale simulations for the period of the Aerosols, Radiation and Clouds in southern Africa (AEROCLO-sA) campaign in September 2017. The study presents a thorough and comprehensive investigation of combined function of tropical temperate troughs (TTTs) and cut-off lows (CoLs) in the formation of the river of smoke. Authors discuss the role of TTTs and CoLs in this context as a novel concept.

The study looks sound, and the paper reads well. I have a few questions and suggestions for possible improvements before publication.

**Specific comments:**

1) This paper investigates a mass transport feature referred to as the river of smoke, thus the title "Smoke in the river" is somewhat misleading. Also, it might be useful to convey the key features of the article, the function of TTTs and CoLs in the formation of the river of smoke, in the title. This might be useful for the searchability of the article, as this is a novel and interesting concept. Further, and to a lesser extent, the use of the acronym AEROCLO-sA in the title can be confusing for the unfamiliar reader. I suggest a possible revision of the title.

Regarding the inclusion of AEROCLO-sA in the title, we believe this is not an issue since the paper will be part of a special collection meant to publicize results from several field campaigns that took place over the southeastern Atlantic in the period 2016-2018: ORACLES, CLARIFY, AEROCLO-sA, LASIC, ….

Regarding the first part of the title, it was more of twist, a word play for those of us authors having grown up in the 70s with one of the most famous rock anthem, Deep Purple's "Smoke on the Water".

We initially tried to squeeze in the title TTT and CoL, but it did not sound as good ;-) Furthermore the current title is also likely to spark curiosity from the community…

Regarding searchability… Hey! that is what key words are for…

We appreciate your thoughtful comments on this… but we would like very much keeping the title as is.

2) This study makes use of high-resolution (5 km) model simulations with Meso-NH and discusses model output in terms of investigating the atmospheric composition and locating the BBA in accordance with observational data. Given the high-resolution of the model and sharp gradients in BBA on the boundaries of the river of smoke, model transport scheme and local mass conservation can have an impact on the results. It has been shown that the inappropriate choice of mass conservation schemes (e.g., global schemes for local case studies) can result in the erroneous creation/transport of mass locally and the use of local mass conservation schemes (e.g. ILMC) can improve the model performance locally and in the presence of sharp mass/concentration gradients (Sørensen et al., 2013;

de Grandpré et al., 2016; Fathi et al., 2021). Please comment on and/or provide reference(s) for the performance of the high-res model setup employed in this study in terms of mass conservation and transport schemes.

We agree on the importance of the transport schemes for the model performance. We have added information and references related to the Meso-NH performances on the matter in the revised MS in Section 2.1.2, as:

"Wind is advected using a fourth-order centered scheme coupled to an explicit fourth-order centered Runge-Kutta time splitting (Lunet et al., 2017) and the other variables are advected with the piece-wise parabolic method (PPM) advection scheme (Colella and Woodward, 1984), a scheme with excellent mass-conservation properties and low numerical diffusion (Müller, 1992)."

Colella, P. and Woodward, P. R.: The Piecewise Parabolic Method (PPM) for gas-dynamical simulations, J. Comput. Phys., 54, 174–201, https://doi.org/10.1016/0021-9991(84)90143-8, 1984.

Lunet, T., Lac, C., Auguste, F., Visentin, F., Masson, V., and Escobar, J.: Combination of WENO and Explicit Runge–Kutta Methods for Wind Transport in the Meso-NH Model, Monthly Weather Review, 145, 3817–3838, https://doi.org/10.1175/MWR-D-16-0343.1, 2017.

Müller, R.: The performance of classical versus modern finite-volume advection schemes for atmospheric modelling in a one-dimensional test-bed, Mon. Weather Rev., 120, 1407–1415, https://doi.org/10.1175/1520-0493(1992)120<1407:TPOCVM>2.0.CO;2, 1992.

3) Considering the impacts of transport and distribution of BBA related aerosols and gases (river of smoke) for example "potential important implications for the radiative and the marine productivity of the region" as mentioned in the introduction section of the article, is there value in estimating the aerosol/gas mass-flux using downwind aircraft measurements (Peischl et al.,2010; Gordon et al., 2015; Fathi et al., 2021). Are Dropsonde and Lidar measurement data useful for mass-flux calculations and has this been attempted in the context of AEROCLO-sA? Please comment.

We agree with the reviewer that estimating the aerosol/gas mass-flux associated with the rivers of smoke is important, in particular for assessing the impact on marine productivity. Such a study has not yet been conducted using the AEROCLO-sA aircraft lidar and dropsonde datasets, the main reason being that lidar observations cannot be used to derive information on aerosol or gas concentration or mass without considerable approximations and assumptions. Furthermore, the flight plans were not really optimized to conduct such an estimation. Nevertheless, our airborne datasets can be used indirectly in a model-based estimation of the aerosol/gas mass-flux using the Meso-NH simulations. As a matter of fact, lidar and dropsonde observations are instrumental in conducting atmospheric dynamics-, thermodynamics- and composition-related model verifications that are needed to derive realistic aerosol/gas mass-flux estimations. Such a study using Meso-NH was conducted for instance by Bou Karam et al. (2009) to estimate Sahelian dust emissions.

Bou Karam, D., C. Flamant, P. Tulet, J.-P. Chaboureau, A. Dabas, and M. C. Todd, 2009: Estimate of Sahelian dust emissions in the intertropical discontinuity region of the West African Monsoon, J. Geophys. Res., 114, D13106, doi:10.1029/2008JD011444.

4) Line 174: At what horizontal spacing and frequency were the dropsondes released? Any relevant limitations?

Dropsondes are expensive (~850 €) and their use should serve a specific research purpose like the documentation of the dynamics and thermodynamics characteristics of the lower troposphere in a given area or at a given time. Hence, during the flights reported here, the horizontal spacing and frequency were imposed by the scientific goals pursued.

However, there is a limitation in the number of dropsondes that can be released during a given 3 h flight. The Vaisala AVAPS system used to track the dropsondes during their descent only has four channels, meaning that the only four dropsondes can be monitored at any given time. Given that it

takes ~20 min for a dropsonde released from a 10 km altitude to reach the surface, a fresh set of dropsondes can only be launched every ~30 min, taking into account the time needed by the operator to initialize the dropsondes. We consider that given the numerous manipulations that have to be conducted by the dropsonde operator, and because the operator also has other tasks during the flight, a maximum of 10 dropsondes can be released during a 3 h Flacon 20 flight.

5) Lines 181-183: Regarding the two flights (F06 and F09), is the choice of flight path important (e.g., counter clockwise, clockwise)? If it is important, please explain why.

The flight paths during the AEROCLO-sA campaign aimed at characterizing BBA- or dust-related emission or transport processes over specific areas or at a particular time of day. This is basically what dictated the clockwise or counter-clockwise direction of the flights.

During flight F06 we were targeting dust emissions processes over the Etosha pan, and the subsequent transport of emitted dust towards the Atlantic Ocean. The counter clockwise direction of the flight was imposed by the fact that dust emissions over Etosha occur shortly after sunrise and that we need to get there at the very beginning of the flight. Then, as the average near-surface wind was northeasterly over the Namibian plateau, the aircraft flight track was designed to follow the main dust transport path.

During Flight F09 we were targeting the characterization of the BBA plume, both remotely and in situ, and the evolution of these characteristics away from the main fire regions in Angola. We also aimed at investigating the impact of the 'brighter than surrounding' surface associated with the Etosha salt pan on the BBA radiative forcing near maximum insolation, hence at the beginning of the flight. These are the reasons behind the clockwise direction of the flight.

This information has been added in the revised version of the MS, as:

"The flight path during the AEROCLO-sA campaign aimed at characterizing BBA- or dust-related emission or transport processes over specific areas or at a particular time of day. The clockwise or counter-clockwise orientation of the flights were therefore dictated by the need to adapt to the specific emission and transport conditions expected for each flight."

6) Line 536: Regarding "… thick clouds embedded in the river of smoke", are there any possible interactions with BBA in terms of cloud formation (e.g., nucleation)?

This is an excellent point made by the reviewer here. Indeed, Hennigan et al. (2021) discuss how the production of secondary organic aerosol in BBA plumes significantly enhance cloud condensation nuclei (CCN) concentrations and how global model simulations predict that nucleation in photo-chemically aging fire plumes produces dramatically higher CCN concentrations over widespread areas of the southern hemisphere during the dry, burning season (Sept.–Oct.).

This information has been added in the revised version of the MS, as:

"For instance, Hennigan et al. (2021) discuss how the production of secondary organic aerosol in BBA plumes significantly enhance cloud condensation nuclei concentrations and how global model simulations predict that nucleation in photo-chemically aging fire plumes produces dramatically higher cloud condensation nuclei concentrations over widespread areas of the southern hemisphere during the dry, burning season (September–October)."

Hennigan, C. J., D. M. Westervelt, I. Riipinen, G. J. Engelhart, T. Lee, J. L. Collett Jr., S. N. Pandis, P. J. Adams, and A. L. Robinson, 2012: New particle formation and growth in biomass burning plumes: An important source of cloud condensation nuclei, Geophys. Res. Lett.,39, L09805, doi:10.1029/2012GL050930

7) Lastly, a general comment on article structure. Results and discussions are presented simultaneously in three different sections (3, 4 and 5). It might be useful, in terms of readability, if these were grouped

together under a results and discussion section. However, this is just a minor suggestion; the article is clear and easy to follow as is.

We made an attempt to comply with the reviewer's comment but were not satisfied by the result. Hence we decided to keep the current narrative.

**Technical corrections:**

1. Line 24: revise "temperate tropical trough" to "tropical temperate trough (TTT)" to be consistent with the rest of the text.

   Corrected. Thanks

2. Line 65: cite the final published version of Gaetani et al., 2021

   Done

3. Line 108: TTL or TTT?

   TTT, corrected.

4. Line 117: acronym ECMWF is not written out in full anywhere in the manuscript

   Corrected.

5. Line 226: Figure 2, subplot details (e.g., contour labels) are very hard to read (small).

   Subplot details in Figure 2 have been improved.

6. Line 234: Do you mean Potential Vorticity (PV)?

   Yes, thanks. Corrected.

7. Line 258: revise "… area of interest in under …" to "… area of interest is under …"

   Corrected

8. Line 271: Do you mean Figure3d?

   Yes, indeed. Corrected.

9. Line 274: ~20°W or ~20°E?

   Yes, indeed. Corrected

10. Line 340: Do you mean "west of the continent"?

    We meant 'west of the continent, over the ocean'

11. Lines 388,396: Figure 9, wrong figure panel labels are used. References to Figure 9 are generally confusing. Figure 9 shows a height-distance cross-section while the text makes time and geographical references (e.g., morning, afternoon, northern part of the flight). It would be useful if flight legs (e.g., north, south, west, east) were labeled on the graph along the distance axis and a few reference timestamps were also provided for the same.

    Indeed, the reference to morning and afternoon was confusing. Mention to 'morning' and 'afternoon' were removed as the flight on 5 September took place in the morning and the flight on 6 September took place in the afternoon.

12. Line 424: Do you mean Figure S5?

    Yes, indeed. Figure 5a.

13. Line 481: Despite being in the title, there is no mention of TTT in section 5?

    Correct. TTT was removed from the title of Section 5.

14. Line 492: Revise to "The time-height cross-section of PV …"

Revised.

15. Line 525: Please provide figure/text reference for "… the airborne lidar measurements in the area of Windpoort."

Information has been added in the form "(~900 km into the flight in Figure 9a and ~250 km into the flight in Figure 9b)."

16. Line 529: Please provide figure/text reference for "…, in accordance with the lidar observations"

Information has been added in the form "(~400 km and 850 km into flight #6 in Figure 9a)."

17. Line 571: Figure 15 was never mentioned before this line, it is not common to introduce new figures in the conclusion section.

Figure 15 is a schematic summarizing the evolution of the situation leading to the formation of the river of smoke. We have renamed Section 6 as 'Summary and conclusions" to account for the fact that Section 6 also includes a summary of the study.

18. Figure 4: End of first line in the caption, do you mean "… at 1200 UTC"?

Yes, indeed. Thanks.

19. Figure 5: Second last line in the caption, do you mean "… geopotential at 300 hPa is a tidal effect…"? The periodic trend is observable in the 300 hPa curve (blue) more prominently!

No, the tidal effect is seen 700 hPa, it is a sub-diurnal cycle.

20. Figure 6: Panels don't seem to have labels (a,b,c,d,e)

Corrected.

21. Figure 8: Details are hard to read.

This Figure will be provided as high resolution for the production stage.

22. Figure 9: Panel labels don't match the figure caption and the article text.

Thanks. This is now corrected.

23. Figure 11: Why are wind data missing?

This is due to data loss in the dropsonde data. Winds from the dropsondes are computed using GNSS satellites. When not enough satellites are 'visible' by the sonde (at least 4), then the is now wind measurements.

24. Figure 13: No panel labels provided?

Corrected.

25. Figure 14: No scale is provided for the contours, maybe label the contours!

We have added information in the revised MS. The caption of Figure 14 now is: "Blue contours represent liquid water every 0.1 g kg$^{-1}$ while brown contours represent potential vorticity every -1.5 PVU (potential vorticity unit, PVU = $10^{-6}$ K kg$^{-1}$ m$^2$ s$^{-1}$)."

26. Figure S5: Repeated panel labels (a,b) in the right column!

Thanks. Corrected.

27. Line 744: The hyperlink doesn't seem to be valid.

**References**

de Grandpré, J., Tanguay, M., Qaddouri, A., Zerroukat, M., and McLinden, C. A.: Semi-Lagrangian Advection of Stratospheric Ozone on a Yin–Yang Grid System, Mon. Weather Rev., 144, 1035–1050, https://doi.org/10.1175/MWR-D-15-0142.1, 2016.

Fathi, S., Gordon, M., Makar, P. A., Akingunola, A., Darlington, A., Liggio, J., Hayden, K., and Li, S.-M.: Evaluating the impact of storage-and-release on aircraft-based mass-balance methodology using a regional air-quality model, Atmos. Chem. Phys., 21, 15461–15491, https://doi.org/10.5194/acp-21-15461-2021 , 2021.

Gordon, M., Li, S.-M., Staebler, R., Darlington, A., Hayden, K., O'Brien, J., and Wolde, M.: Determining air pollutant emission rates based on mass balance using airborne measurement data over the Alberta oil sands operations, Atmos. Meas. Tech., 8, 3745–3765, https://doi.org/10.5194/amt-8-3745-2015, 2015.

Peischl, J., Ryerson, T. B., Holloway, J. S., Parrish, D. D., Trainer, M., Frost, G. J., Aikin, K. C., Brown, S. S., Dubé, W. P., Stark, H., and Fehsenfeld, F. C.: A top-down analysis of emissions from selected Texas power plants during TexAQS 2000 and 2006, J. Geophys. Res.-Atmos., 115, D16303, https://doi.org/10.1029/2009JD013527, 2010.

Sørensen, B., Kaas, E., and Korsholm, U. S.: A mass-conserving and multi-tracer efficient transport scheme in the online integrated Enviro-HIRLAM model, Geosci. Model Dev., 6, 1029–1042, https://doi.org/10.5194/gmd-6-1029-2013, 2013.

---

## Author Comment (AC2)

**ACP-2021-939: Reply to Reviewer 2**

**Smoke in the river: an AEROCLO-sA case study, Cyrille Flamant et al.**

The authors would like to thank the reviewer for her/his valuable comments which helped improving the quality of the manuscript. Our point-by-point responses to the reviewer's comments appear in blue below. The text modified in the revised version of the MS and included in the response appears in red.

**General comments:**

The manuscript entitled "Smoke in the river: an AEROCLO-sA case study" written by Cyrille Flament presented the formation of a river of smoke over south Africa found in AEROCLO-sA campaign. Based on full dataset of reanalyses data, numerical modeling, ground-based, airborne, and space-borne measurements, this study suggested the interaction between temperate tropical trough (TTT) and cut-off low (CoL) to promote the transport of biomass burning aerosols. This kind of study is essential to interpret "smoke in the river" and I would like to consider the possible publication. However, I have fundamental questions on the numerical modeling used in this study. I would like to request to address my concerns listed below.

**Major comments:**

Description of Meso-NH: The current description of Meso-NH includes ambiguous statement. Please clarify following specific points.

Why high resolution simulation is needed in this study?

Using fine-scale Meso-NH simulation allows use to conduct direct comparisons with high resolution airborne observations at a commensurate scale (e.g. 1 km horizontal resolution and 30 m vertical resolution for the LNG lidar data). Furthermore, while the CAMS and ERA5 products are suitable for describing the large-scale circulation and aerosol distribution, high resolution modelling allows having a better description of the topography in the area and its influence on aerosol transport, especially in the presence of escarpments as is the case along the Namibian coastline. In section 2.1.2, we added:

"Running the model at a relatively fine resolution allows a comparison with high spatio-temporal resolution airborne observations at a commensurate scale, as well as having a better description of the topography in the area and its influence on aerosol transport."

As stated in the MS, there are 64 levels in the Meso-NH simulations, including 14 in the lower 1 km and 30 in the lower 6 km of the atmosphere, with bins sizes ranging from 60 m near the surface to 600 m above 7 km. In section 2.1.2, we added (line 149):

"…(14 of which in the lower 1 km and 30 in the lower 6 km)…"

For the analysis from 2 September, there is only one day spin-up time. Is it appropriate to adequately remove the initial condition? (Even in Figure 14, 1 September is plotted but is it appropriate?) Related to this point, there is no description for the initial condition data. What kind of initial condition was assumed?

We do not use initial condition for aerosols, so basically the BB tracer concentrations are set to 0 at the beginning of the simulation and the BB tracer concentrations build-up with time. It can be hypothesized that, in source regions, BB tracers will fill the lower 6 km of the troposphere in about 12 h, i.e. approximately the time for the atmospheric boundary layer to develop during the day.

Therefore, the assessment of the BBA Simulation is meaningful only after a spin-up period that we estimate at 12 hours. This is now mentioned in the MS in Section 2.1.2:

"No initial conditions are assumed for BBA, so that BB trace concentrations builds up with time in the simulation domain. In source regions, BB tracers will fill the lower 6 km of the troposphere in about 12 h, i.e. approximately the time for the atmospheric boundary layer to develop during the day. Therefore, the assessment of the BBA Simulation is meaningful only after a spin-up period that we estimate at 12 hours."

We agree that showing BB tracer related AOT on 1 September can be deceptive. Hence, as suggested by the reviewer, we have excluded 1 September from the comparison with AERONET in Fig. 14 and with now are showing Fig 14 starting on 2 September.

[Figure]

The proxy for BBA are used by organic matter taken from CAMS whereas black carbon is analyzed from Meso-NH. I guess that black carbon is an important proxy to represent BBA, but the purpose of using black carbon is passive tracer?

Agreed. The expression 'black carbon passive tracer' is misleading. In fact, we use in Meso-NH a BB tracer that includes both organic carbon and black carbon. GFED emission maps are based on BB

carbon fluxes, including both black and organic carbon. This is now corrected throughout the MS and in the figures.

Only GFED emission is used to calculate Meso-NH chemistry? Even though the biomass burning is dominant emission sources to this analysis, available emissions of anthropogenic source and biogenic source are needed to be considered to represent the chemical field over modeling domain.

This study is not focused on the chemistry of smoke, and reactive chemistry is beyond the scope of the paper, which focuses on meteorology and atmospheric dynamics. For the period of interest, the CAMS simulations highlight that BBA dominate the atmospheric composition, and other sources (pollution, biogenic, traffic) do not significantly contribute to the aerosol load.

As stated, GFED emission's grid resolution is 0.25 degree and this is approximately five times greater than the grid resolution of Meso-NH. How to interpolate into 5 km horizontal spacing? Without the fine-scale representation of emission itself, what is the advantage to conduct fine-scale Meso-NH simulation?

Here, we use a standard interpolation on spatial grid. Regarding the resolution, see our answer at point #1

In addition, because the treatment of vertical allocation is an important aspect to describe biomass burning emission sources, the information of the vertical grid allocation is required to understand the modeling treatment.

In the case of pyroconvection, it is indeed correct that the vertical allocation of BBA could be an issue. However, this is not the case over the region considered here as already discussed in several studies (e.g. Labonne et al., 2017; Menut et al., 2018; Mallet et al., 2020, among others). On average, fires are not intense enough to inject BBA above atmospheric boundary layer.

In section 2.1.2, we added:

"The BB tracer is then mixed vertically by turbulence in the atmospheric boundary layer. Fires in the area of interest are not intense enough to inject BBA above the atmospheric boundary layer as discussed in several studies (e.g. Labonne et al., 2017; Menut et al., 2018; Mallet et al., 2020, among others)."

Labonne, M., Breon, F.-M., and Chevallier, F.: Injection height of biomass burning aerosols as seen from a spaceborne lidar, Geophys. Res. Lett., 34, L11806, https://doi.org/10.1029/2007GL029311, 2007.

M. Mallet, F. Solmon, P. Nabat, N. Elguindi, F. Waquet, D. Bouniol, A. Sayer, K. Mayer, R. Roehrig, M. Michou, P. Zuidema, C. Flamant, J. Redemann, and P. Formenti, 2020: Direct and semi-direct radiative forcing of biomass burning aerosols over the Southeast Atlantic (SEA) and its sensitivity to absorbing properties: a regional climate modeling study, *Atmos. Chem. Phys.*, **20,** 13191–13216.

L. Menut, C. Flamant, S. Turquety, A. Deroubaix, P. Chazette and R. Meynadier, 2018: Impact of biomass burning on pollutants surface concentrations in megacities of the Gulf of Guinea, *Atmos. Chem. Phys.* 18, 2687–2707.

After the clarification of these questions on model configuration, I am wondering the performance of Meso-NH itself. There is no direct comparison for modeled black carbon. In Figure 9, model were indirectly compared to measured attenuated backscatter coefficient. Despite the discussion in line 396-404, I simply impressed that model posed much mixing of black carbon from the surface to 6 km.

Observed high coefficients were only found in 3-6 km, and I am suspicious the modeling skill to capture the measurement. Without the appropriate modeling performance, results drawn from Meso-NH model could be also suspicious. I would like to request to include more discussion to reinforce the modeling performance by Meso-NNH to represent the behavior of black carbon.

In Figure 9, we are now showing comparison between extinction derived from the airborne LNG lidar and extinction derived from Meso-NH at 1604 nm. This allows a more direct comparison between observations and model outputs. We have added a description of the lidar-derived extinction products in Section 2.2.3:

"Extinction coefficients at 1064 nm are retrieved from ABC profiles using a standard lidar inversion method that employs a lidar ratio of 40 sr, characteristic of BBA. The retrievals have an estimated uncertainty of 15 %."

The new Figure 9 showing extinction coefficients instead of ABC is shown below.

[Figure]

**Specific comments:**

Line 24: This is not consistent to line 81. Please confirm.

Agreed. For the sake of consistency and accounting for the referee's comment, we have modified the sentence line 81 as:

"Tropical temperate troughs (TTTs) typically form when a tropical disturbance in the lower atmosphere is coupled with a mid-latitude trough in the upper atmosphere (Lyons, 1991)."

Line 94: The campaign period was August-September, and the analyzed event was early September. In my understanding, this analyzed period could be regarded as late winter to early spring and not to fit winter. Is this contradict to the statement in Line 91-92?

Agreed. The period of the campaign corresponds to late winter *stricto sensus*. We have modified the sentence line 91-92 to reflect this as:

"However, the role of TTTs in the transport of BBA during the **late** winter has never been investigated until now."

Line 108: There is no definition for "TTL". Please clarify.

Corrected, should read TTT.

Line 150: If authors use four-digit as HHHH, "000 UTC" should be "0000 UTC".

Corrected.

Line 169: Please confirm the wavelength of AERONET dataset. If 500 nm, this is slightly different to calculated AOT by Meso-NH. How can we understand the difference of wavelength in the comparison between calculation and observation?

We indeed use AERONET AOT data at 500 m. There are countless examples in the literature of AOT comparison at different, but close wavelength, as is the case here. Nevertheless we can make a back of the envelop estimate of the error/uncertainty associated with the wavelength difference, based on the fact that the extinction (and hence AOT) varies with wavelength according to a power law, $Ext\_550/Ext\_500=(550/500)^a$, with $a\sim1$ between 450 and 660 nm as assessed by Denjean et al. (2020) for elevated BBA coming from Central Africa. As a result, the error/uncertainty induced by the difference of wavelength is on the order of 10%. Moreover, Gaetani et al. (2021, see Figure 3) compared the daily AOT at AERONET stations with the CAMS product at 550 nm on a several year span in southern Africa (2003-2017) and found a robust linear relationship between the ln(AOT) at 500 and 550 nm, indicating that the AOT at 550 nm can be used a reliable proxy for the AOT at 500 nm. This is now discussed in the caption of Figure 8, where the first comparison between the AERONET AOT at 500 nm and CAMS and MODIS products at 550 nm is made:

"A robust linear relationship is observed between the natural logarithm of the AOT at 500 and 550 nm (see Denjean et al., 2020; and Gaetani et al., 2021), therefore CAMS and MODIS products at 550 nm can be used as reliable proxies for the AOT at 500 nm at the AERONET stations".

Denjean, C., Brito, J., Libois, Q., Mallet, M., Bourrianne, T., Burnet, F., et al. (2020). Unexpected biomass burning aerosol absorption enhancement explained by black carbon mixing state. Geophysical Research Letters, 47, e2020GL089055. https://doi.org/10.1029/2020GL089055

Gaetani, M., Pohl, B., Alvarez Castro, M. C., Flamant, C., and Formenti, P.: A weather regime characterisation of winter biomass aerosol transport from southern Africa, Atmos. Chem. Phys., 21, 16575-16591, https://doi.org/10.5194/acp-21-16575-2021, 2021.

Line 389: Does "morning" mean small distance because this flight started from 0736 UTC? Sometimes it is ambiguous to use time using this Figure 9, so it would be better to add another x-axis represented by time.

Agreed, this is ambiguous indeed as the whole flight takes place from 0736 to 1014 UTC. We have modified the sentence as: "On 5 September, the BBA layer is observed…"

Line 571-572: I understand that Figure 15 summarizes and illustrated the finding in this study. However, this should be fully discussed before the conclusion section. Please move these discussions related to Figure 15 into Section 5.

Thanks for the suggestion. However, we have decided to keep the Figure where it was, as Section 5 is not a summary section, unlike Section 6. Nevertheless, to comply with the reviewer's comment we have renamed Section 6 as 'Summary and conclusions".

Figure 2: The characters for mean sea pressure level is small. Please enlarge, or use color-scale to distinguish them.

Readability of Figure 2 has been improved.

Figures 3 and 4: The expression of data stated in the caption should be unified through manuscript. "1 September" is used in Figure 2, but "01/09" is used in Figure 3. These are hard to read.

OK, agreed. Caption have been homogenized through the manuscript as xx September, and hence modified in the caption of Figures 3 and 4. The captions now read:

Figure 6: There is no indexes to represent (a) to (f) within this figure. In the caption, MODIS is repeated. Please rephrase.

Figure corrected. The caption is now corrected

Figure 8: Please enhance the black color. This seems as gray color and hard to read.

The contrast is fine in the ps and pdf format of the figure.

Figures 10 and 12: Because four panels are not unified as altitude levels, it is hard to follow the meaning of this figure. It is much straightforward to align as 1-2, 2-3, 4-5, 6-7 km (or vice versa). Please reconsider the order of panel and also rearrange the discussion main text.

The logic behind the ordering of the sub-figures was more to have the upper level trajectories on the top panels and the lower level trajectories in the bottom panels.

Figure 13: There is no indexes to represent (a) to (d) within this figure. Please move the panel of vertical velocity because only this panel is slightly differently positioned.

Labels have been added and panel positions corrected.

Figure 14: What is the contours level for liquid water (blue) and potential vorticity (brown)?

We have added information in the revised MS. The caption of Figure 14 now includes the information as: "Blue contours represent liquid water every 0.1 g kg$^{-1}$ while brown contours represent potential vorticity every -1.5 PVU (potential vorticity unit, PVU = $10^{-6}$ K kg$^{-1}$ m$^2$ s$^{-1}$)."